# Stochastic priming and spatial cues orchestrate heterogeneous clonal contribution to mouse pancreas organogenesis

Hjalte List Larsen[1], Laura Martín-Coll[1], Alexander Valentin Nielsen[2], Christopher V. E. Wright[3], Ala Trusina[2], Yung Hae Kim[1] & Anne Grapin-Botton [1]

Spatiotemporal balancing of cellular proliferation and differentiation is crucial for postnatal tissue homoeostasis and organogenesis. During embryonic development, pancreatic progenitors simultaneously proliferate and differentiate into the endocrine, ductal and acinar lineages. Using in vivo clonal analysis in the founder population of the pancreas here we reveal highly heterogeneous contribution of single progenitors to organ formation. While some progenitors are bona fide multipotent and contribute progeny to all major pancreatic cell lineages, we also identify numerous unipotent endocrine and ducto-endocrine bipotent clones. Single-cell transcriptional profiling at E9.5 reveals that endocrine-committed cells are molecularly distinct, whereas multipotent and bipotent progenitors do not exhibit different expression profiles. Clone size and composition support a probabilistic model of cell fate allocation and in silico simulations predict a transient wave of acinar differentiation around E11.5, while endocrine differentiation is proportionally decreased. Increased proliferative capacity of outer progenitors is further proposed to impact clonal expansion.

[1] DanStem, University of Copenhagen, 3B Blegdamsvej, DK-2200 Copenhagen N, Denmark. [2] Niels Bohr Institute, University of Copenhagen, 17 Blegdamsvej, DK-2200 Copenhagen N, Denmark. [3] Department of Cell & Developmental Biology, Vanderbilt University, Nashville, TN 37232-0494, USA. Correspondence and requests for materials should be addressed to Y.H.K. (email: yung.kim@sund.ku.dk) or to A.G.-B. (email: anne.grapin-botton@sund.ku.dk)

Defining the rules governing embryonic organ development and postnatal tissue homoeostasis is essential for understanding disease pathology and for the generation of functional cell types for regenerative medicine purposes. Seminal studies have demonstrated how rapidly proliferating postnatal tissues such as the skin and the intestine are homeostatically maintained by equipotent stem cells undergoing seemingly stochastic cell fates choices by neutral competition for limited niche signals[1–4]. In contrast to postnatal tissue homoeostasis, embryonic development of most organs occurs at a state of system disequilibrium, as a population of progenitors expands while simultaneously giving rise to differentiating progeny. Although optimality in the design of strategies ensuring rapid organ development has been proposed[5], little is known regarding how global embryonic organogenesis is orchestrated when deconstructed into clonal units originating from single progenitors at the onset of organ bud formation. Studies of retinal development have provided compelling evidence for a stochastic process of cell fate choices using both in vitro[6] and in vivo approaches[7]. However, a deterministic model of embryonic neocortical development was proposed[8], based on the observation of similar behaviour of the two daughters of individual cells. These discrepancies in organ design emphasise the need for studies investigating individual cell progenies in other organ systems. Here we investigate how the allocation of endocrine and acinar fates is balanced with progenitor expansion from the beginning of pancreas formation using clonal analysis and single-cell molecular profiling.

Embryonic mouse pancreas development is initiated at around embryonic day (E)9.0 by the specification of pancreatic progenitors at the dorsal and ventral sides of the posterior foregut endoderm[9]. Though induced by different mechanisms, the two anlage are composed of expanding $Pdx1^+Hnf1b^+Sox9^+Ptf1a^+$ progenitors forming bud-like structures protruding into the surrounding mesenchyme[10]. A small number of $Neurog3^+$ endocrine precursors giving rise to the endocrine lineage of the pancreas are also found in these early buds[11, 12]. Morphogenetic processes occur concomitantly leading to the formation of lumens and their organisation into a plexus and subsequent tree-like branches[13, 14]. While the distal tip-domain is comprised of $Ptf1a^+$ unipotent acinar progenitors after E13.5[15, 16], the $Hnf1b^+$ trunk domain is bipotent and gives rise to endocrine cells, as well as the ductal cells that will eventually line the epithelial network draining acinar digestive enzymes to the duodenum[17–19]. Following specification towards the endocrine lineage, $Neurog3^+$ endocrine precursors delaminate from the epithelial trunk domain to form immature islet clusters that will eventually mature into the endocrine Islets of Langerhans[20]. Although population-based lineage tracing has demonstrated the multipotency of the early pancreatic progenitors by virtue of their capability to give rise to progeny in the three major pancreatic lineages[12, 15–17, 21] (Fig. 1a), no study has addressed the clonal contribution of the proposed multipotent pancreatic progenitors (MPCs) to pancreas organogenesis. One previous clonal analysis indeed restricted its focus on the progeny of single endocrine precursors examining their postnatal expansion[22]. Recent studies have demonstrated that pancreatic trunk progenitors undergo stochastic priming towards the endocrine lineage at mid-gestation[19], and thus we questioned whether there are subpopulations of pancreatic progenitors exhibiting restricted lineage potencies from the onset of embryonic pancreas development or whether progeny from equipotent progenitors undergo stochastic lineage commitment.

In this study, using clonal analysis of E9.5 pancreatic progenitors, when the pancreatic primordium has just been specified, we demonstrate that individual pancreatic progenitors contribute heterogeneously to pancreas organogenesis both in progeny size and fate composition. While some progenitors are multipotent per se, giving rise to acinar, endocrine and ductal progeny, we also demonstrate the existence of bipotent ducto-endocrine and unipotent endocrine cells forming half of the primordium. This population represents cells at different stages of progression on the endocrine differentiation path, including proliferative endocrine-committed cells, and exhibits undetectable to low levels of PTF1A. In contrast, bipotent and multipotent clones do not exhibit different expression profiles, suggesting they are not molecularly distinct cell populations. We show that clonal expansion and fate heterogeneity are compatible with a simple model of probabilistic cell fate acquisition operating downstream of spatially controlled proliferative and fate-biasing patterning cues.

## Results

**Single E9.5 pancreatic cells produce heterogeneous progeny**. To investigate how individual cells among the about 500 cells that have just been specified to found the pancreas contribute to organogenesis, we devised a lineage tracing strategy making use of the $Rosa26^{CreER}$ driver (Fig. 1b). The ubiquitous activity of the $Rosa26$ locus ensures $CreER$ expression throughout the developing embryo and hence also enables non-biased labelling of pancreatic cells[23]. We selected the $mT/mG$[24] reporter over other multicolour reporters to be able to mark the differentiation status of clonal progeny. This required the dosage of very low levels of the active tamoxifen metabolite 4-OH tamoxifen (4-OHTm) to reach labelling of only one cell per pancreatic primordium within the 24 h following injection[25]. The labelling index of 11.8% (20 epithelial clones in 170 embryos) ensured a low risk of labelling two progenitors in the same pancreatic bud as of 1.4% (0.118×0.118). Whole-mount staining of E14.5 pancreata for endocrine (PAX6), progenitors lining the ducts (SOX9) and acinar (CPA1) markers enabled us to determine the fate of labelled GFP$^+$ progeny at E14.5, a stage by which acinar cells are committed (Fig. 1c–h)[12, 15–17]. We observed a large extent of clone size heterogeneity, ranging from single-cell clones to clones consisting of hundreds of cells (Fig. 1h). Single GFP$^+$ cells belonged to the endocrine lineage based on immunoreactivity for PAX6, cell morphology and location outside the pancreatic epithelium in islet-like structures (Fig. 1d, f, g). These single cells are expected to result from labelling non-proliferative endocrine cells, their $Neurog3$-expressing precursors or pancreatic progenitors differentiating directly into the endocrine lineage without dividing. We also observed 2- and 3-cell endocrine clones, suggesting that a labelled endocrine-biased progenitor had undergone a single or two rounds of divisions. In line with the postulated existence of multipotent progenitors based on non-clonal analyses, multipotent clones of 40–250 cells were found, consisting of ductal, endocrine and acinar progeny (Fig. 1h). Moreover we did observe bipotent clones of 6–100 cells harbouring only ductal and endocrine progeny, indicating that not all E9.5 progenitors contribute to the acinar lineage during pancreas organogenesis. However, we did not observe unipotent acinar clones arising from E9.5 progenitors. While confirming the existence of MPCs at the single-cell level, our results reveal heterogeneity in potency and contribution to pancreas organogenesis from single pancreatic cells. Furthermore they uncover the existence of bipotent progenitors as early as E9.5 and that half of the cells in the early pancreatic anlage give rise to solely endocrine progeny, a surprising finding considering that the adult endocrine cells only account for about 1% of the adult organ[26].

**Heterogeneous marker expression in E9.5 progenitors.** Heterogeneity in the clonal progeny may be either due to an intrinsic lineage bias in sub-populations of E9.5 progenitors or

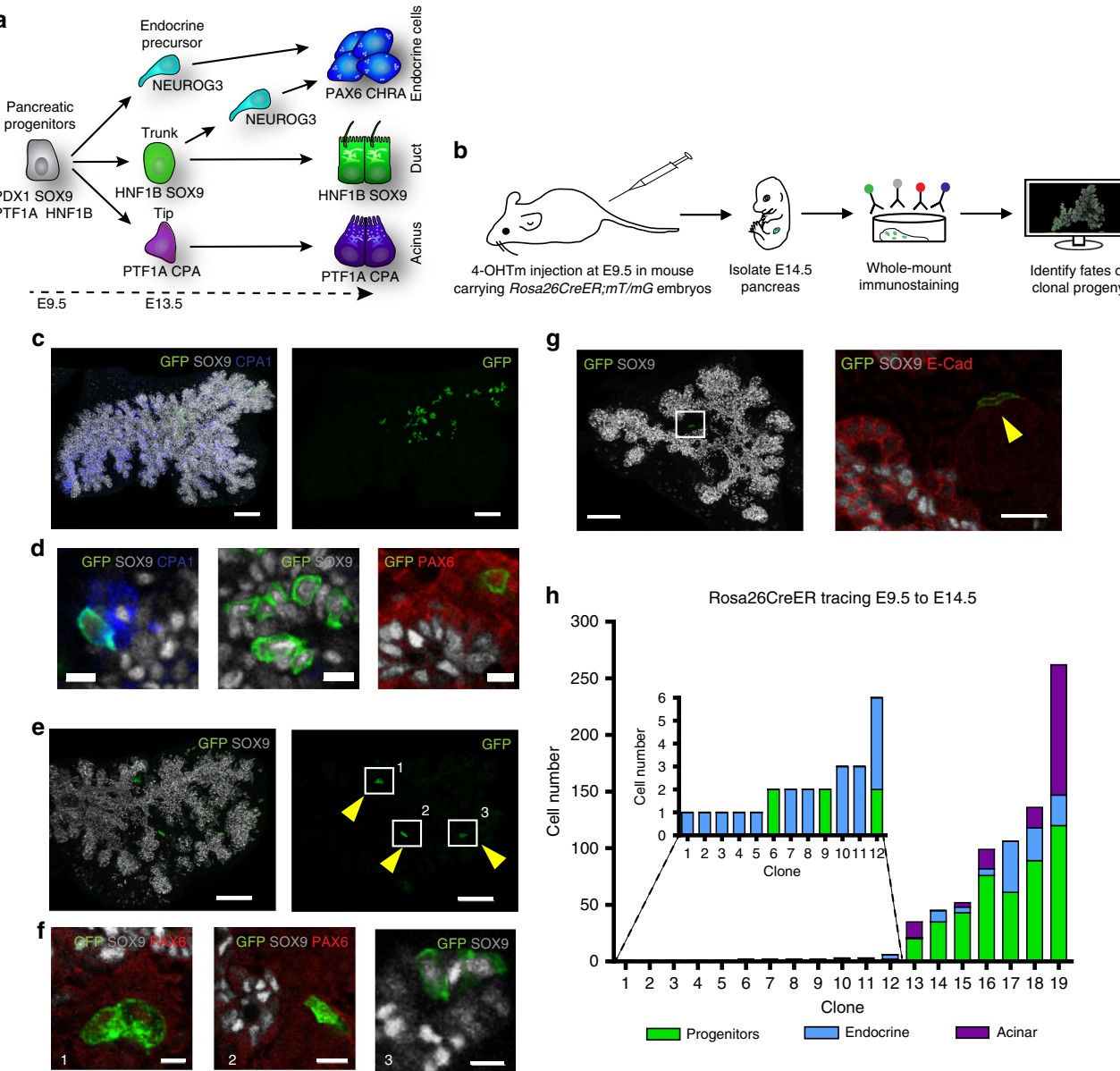

**Fig. 1** *Rosa26^CreER*-mediated clonal analysis reveals heterogeneous contribution of E9.5 progenitors to pancreas organogenesis. **a** Schematic overview of lineage relationships based on previous global lineage tracing. **b** Schematic overview of strategy applied to identify fates of clonal progeny from E9.5 pancreatic progenitors. **c** 3D maximum intensity projection (MIP) of a large, multipotent clone containing acinar (CPA1), progenitors lining the ducts (SOX9) and endocrine progeny (PAX6, from other multipotent clone) **d**. Scale bars, 100 μm **c** and 15 μm **d**. **e** 3D MIP of a bipotent clone containing endocrine (insets 1 and 2) and ductal (inset 3) clonal progeny **f**. Scale bars, 100 μm **e** and 10 μm **f**. **g** 3D MIP and optical section (inset) showing a single-labelled endocrine cell after clonal analysis from E9.5 to E14.5. Note the localisation of the GFP^+ cell in an E-CAD^Low endocrine cluster. Scale bars, 80 μm and 15 μm (inset). **h** Quantification of clone sizes and compositions following clonal analysis from E9.5 to E14.5 (n = 170 embryos, 20 with clones, 1 excluded due to poor immunocytochemistry-the images displayed show representative data from those)

due to the stochastic fate allocation of clonal progeny during progressive organisation and compartmentalisation of the pancreatic epithelium. To test the first hypothesis, we conducted single-cell qRT-PCR following FACS isolation of dorsal foregut progenitor cells at E9.5 (Fig. 2a). tSNE-mediated dimensionality reduction of single cells revealed the existence of three distinct populations (Fig. 2b). On the basis of the expression of known lineage markers, we classify these three clusters as pancreatic endocrine (*Neurog3^+* and *Glucagon^+*), pancreatic progenitors (*Pdx1^+* and *Sox9^+*) and duodenal progenitors (*Cdx2^+*, absence of *Pdx1* and *Sox9*). Interestingly, cells characterised as belonging to the endocrine lineage organised on one projection axis forming a pseudo-temporal trajectory starting from *Neurog3^+* endocrine

precursors and progressing with the expression of markers associated with progressive endocrine maturation (Fig. 2c). These molecularly distinct cells are expected to contribute to the non-proliferative endocrine-committed cells observed in the lineage tracing. When focusing the dimensionality reduction on *Pdx1^+* pancreatic progenitors only, we observed marked heterogeneities in expression of various pancreas-associated transcription factors. Since at E14.5 *Ptf1a* marks acinar cells at the tip while *Nkx6-1*[16, 27], *Hnf1b*[17] and *Hes1*[18] mark bipotent progenitors in the trunk, we investigated whether cells expressing these markers at E9.5 already had specific molecular signatures suggestive of emerging tip and trunk fates. However, in spite of heterogeneous expression of these markers, no global gene

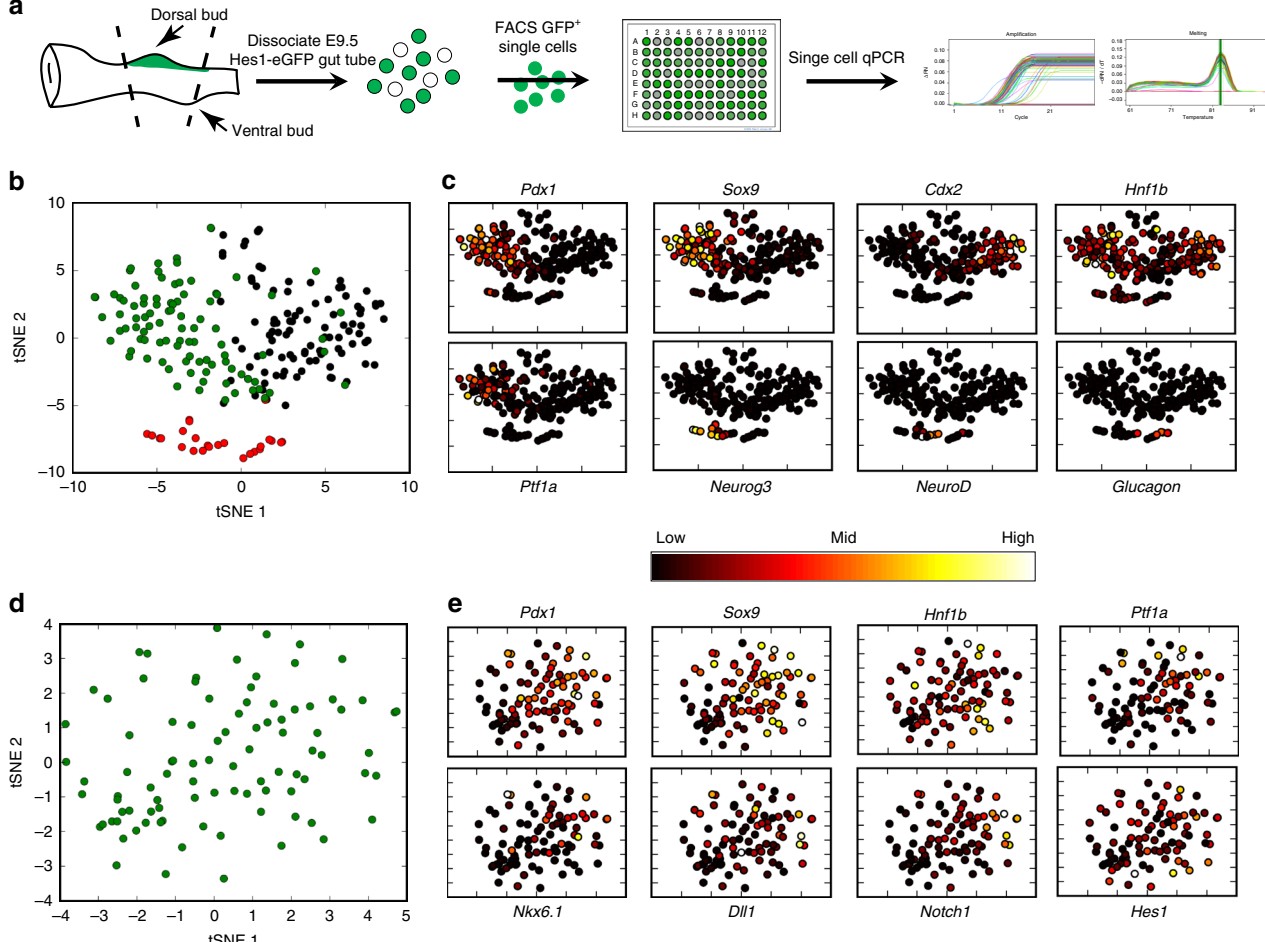

**Fig. 2** Single-cell qRT-PCR of E9.5 pancreatic progenitors reveals endocrine differentiation progression but no apparent tip-trunk segregation signature.
**a** Schematic outline of the strategy used to isolate single pancreatic progenitor cells for single-cell qRT-PCR analysis using the Hes1-eGFP strain[43].
**b** tSNE dimensionality reduction of all single cells passing initial quality controls. Three cell clusters are displayed in *green*, *black* and *red*, corresponding to pancreatic progenitors, duodenal progenitors and pancreatic endocrine cells, respectively. **c** *Hnf1b* is expressed in both pancreatic and duodenal progenitors, whereas *Ptf1a* expression is exclusively detected in pancreatic progenitors albeit in heterogeneous pattern. Cells in the endocrine population cluster organise on a pseudo-temporal differentiation pathway starting with *Neurog3*+ endocrine precursors at the left, progressing into glucagon-expressing terminally differentiated endocrine cells at the right. **d** tSNE dimensionality reduction of *Pdx1*+ pancreatic progenitors only. **e** Despite heterogeneous expression of transcription factors and signalling components involved in subsequent tip-trunk segregation, expression of these markers seems uncorrelated and does not partition *Pdx1*+ progenitors into tip- and trunk-biased clusters

signature was associated with *Nkx6-1*, *Hnf1b* or *Hes1*, and these three markers showed no cross-correlation (Fig. 2d, e; Supplementary Figs. 1–4). Although *Ptf1a* expression did not correlate strongly with specific single markers, *Ptf1a*+-cells clustered after tSNE-mediated dimensionality reduction, suggesting that they are more similar to each other than to other progenitor cells. This molecular analysis uncovers that cells committed to endocrine differentiation can be molecularly identified, whereas subpopulations of multipotent or bipotent progenitors identified by clonal analysis cannot be molecularly predicted with this set of markers.

**Spatial patterns of progenitor marker expression.** To further assess heterogeneity in markers at the protein level, we used whole-mount immunostaining of E9.5 gut tubes and quantitative image analysis (Fig. 3a, b). We observed heterogeneous expression levels of HES1, SOX9 and PTF1A, whereas HNF1B and PDX1 were more homogeneously expressed among progenitors (Fig. 3c, d). Using 3D Voronoi-Delaunay triangulation (Fig. 3a)

and measurements of the correlation in expression levels between neighbouring cells, we observed that HNF1B is expressed at higher levels towards the more posterior side of the pancreatic bud and in the gut tube and that PTF1A expressing cells appear clustered at a medial-bilateral location in E9.5 dorsal buds. Other transcription factors did not show any regionalisation of expression levels (Fig. 3d). These results demonstrate that the transcriptional profiles observed by single-cell qRT-PCR are translated into similar global profiles at the protein level, and that the levels of PTF1A and HNF1B display regionalised patterns in the E9.5 bud.

**Distinct clonal progeny of *Hnf1b*- and *Ptf1a*-expressing cells.** The differential expression of PTF1A and HNF1B at E9.5 and the subsequent segregation of these markers to the tip and trunk domain, respectively, led us to investigate whether single progenitors expressing *Ptf1a* or *Hnf1b* at E9.5 contribute differential progeny by clonal analysis using *Ptf1a*CreER- and *Hnf1b*CreER drivers (Fig. 4a). The *Hnf1b*CreER-driver (labelling

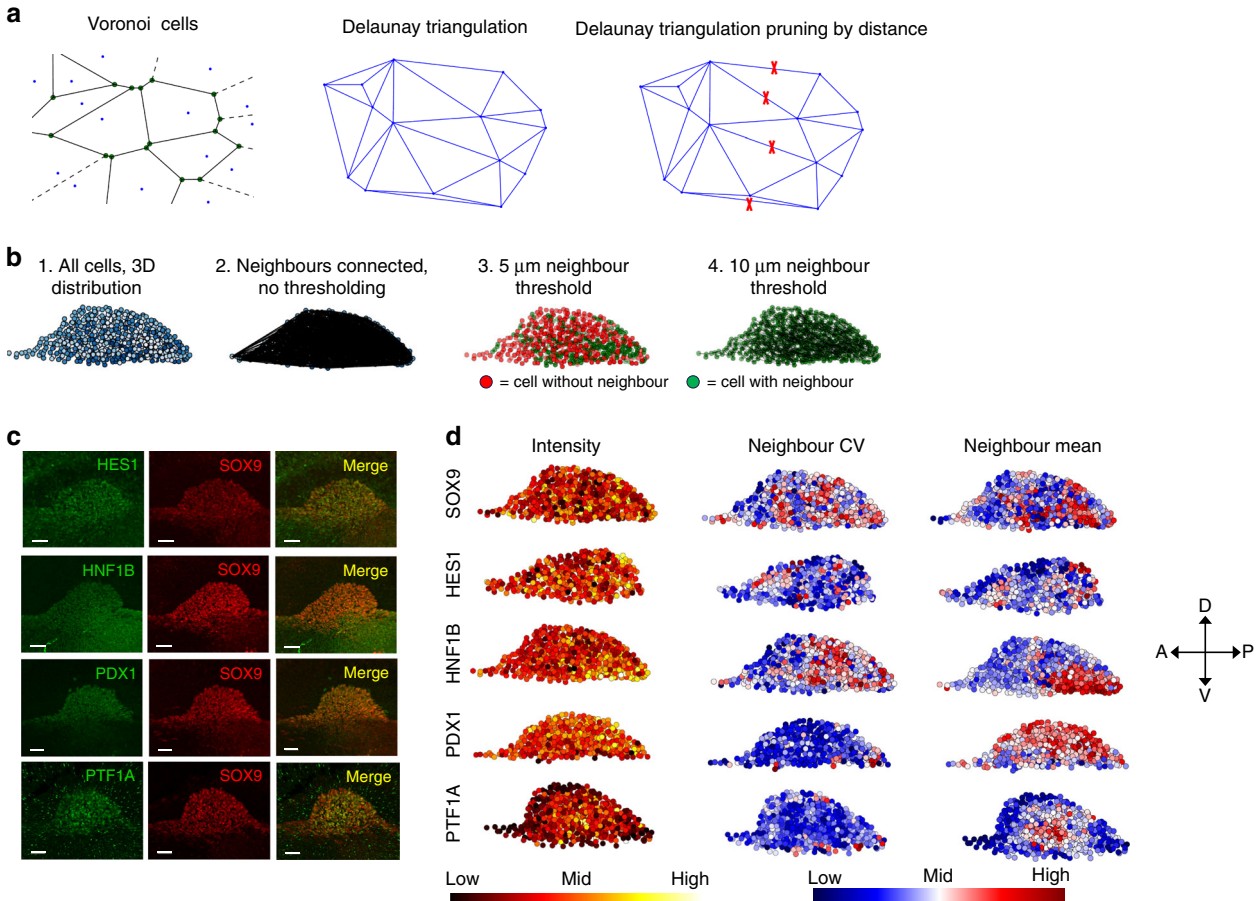

**Fig. 3** E9.5 pancreatic progenitors display heterogeneous expression of pancreas-associated transcription factors and distinct patterns of neighbour cell expression correlation. **a** Schematic of Voronoi-Delaunay triangulation implemented to identify neighbour cells in 3D. Distance-based pruning was implemented in order to identify biologically meaningful neighbours in 3D space. **b** Example of Voronoi-Delaunay triangulation-mediated neighbour identification in E9.5 pancreatic bud. Cells colour-coded according to z-position (1) are subjected to Voronoi-Delaunay triangulation, identifying all neighbour relations in 3D (2). Implementation of a distance threshold of 5 μm (3) or 10 μm (4) generates different number of cells with and without 3D neighbours (*green* and *red* cells, respectively). For all downstream analyses, 10 μm was chosen as neighbour distance threshold. **c** Example of transcription factor expression pattern in E9.5 pancreatic buds following whole-mount staining. 3D MIP is displayed. Scale bars, 30 μm. **d** 3D plots of staining intensity, neighbour coefficient of variation and neighbour mean intensities from immunostaining against the indicated transcription factors. Note the heterogeneous expression patterns of HES1, SOX9 and PTF1A and the regionalised expression of HNF1B (posterior) and PTF1A (lateral) ($n = 2$ for all, except $n = 8$ for SOX9. The images displayed show representative data from those)

index: 35 clones in 120 pancreata, 29%; probability of double labelling, 8%) resulted in a similar pattern of clones as observed with *Rosa26*[CreER], that is unipotent endocrine, bipotent ducto-endocrine as well as multipotent clones (Fig. 4b, c and f). However, the endocrine-committed clones were less frequent, constituting 33% instead of 50% of the total clone repertoire, likely due to the fact that unlike *Rosa26*, *Hnf1b* is not expressed in mature endocrine cells[28]. In addition, we detect HNF1B immunoreactivity in $67.7 \pm 3.8\%$ of the NEUROG3-expressing endocrine precursors at this stage, while *Rosa26* is expected to be expressed in all (Supplementary Fig. 5). A similar frequency of endocrine-committed precursors was observed when tracing *Hnf1b*[CreER]-labelled cells from E9.5 to E10.5 (Supplementary Fig. 6). Interestingly, we observed a clone consisting solely of 6 endocrine cells. Combined with the two 3-cell clones seen in the *Rosa26* tracing, this suggests that some endocrine-biased progenitors can undergo multiple rounds of divisions (Fig. 4b, f; clone # 12), in line with the recent observation that cells with low levels of *Neurog3* transcription can proliferate[29]. The low-differentiation rate towards the endocrine lineage ($p = 0.12 \pm 0.007$, Supplementary Fig. 7) makes independent probabilistic entry of progeny into the endocrine lineage highly

unlikely, suggesting that the E9.5 pancreatic bud contains progenitors biased towards multigenerational endocrine specification. On the other hand, we never observed unipotent acinar clones arising from E9.5 progenitors. Furthermore, acinar-containing clones always contained ductal and endocrine progeny, suggesting that acinar-lineage allocation has not yet occurred in any cell at E9.5. The proportion of multipotent clones was similar to what was observed using the *Rosa26*[CreER] driver after correction for the absence of labelling of mature endocrine cells by *Hnf1b*[CreER]. Though only five of the clones were found in the ventral pancreas, which is smaller than the dorsal, they were bi- or multipotent but not endocrine committed, possibly due to a delay in endocrine program onset in the ventral pancreas. This suggests that the assumed labelling of progenitors with high-expression levels of *Hnf1b* expression at E9.5 does not bias lineage contribution to the trunk domain (Fig. 4f). Similarly, *Ptf1a*[CreER]-based lineage tracing did not bias progenitors towards the acinar lineage, either (Fig. 4d, e and g). However, cells traced by *Ptf1a*[CreER] (labelling index: 13 clones in 30 dorsal pancreata, 44%; probability of double labelling, 19%) did not form endocrine-only clones, unlike what was seen with *Rosa26*[CreER]- and *Hnf1b*[CreER]-drivers. This suggests that cells exhibiting high

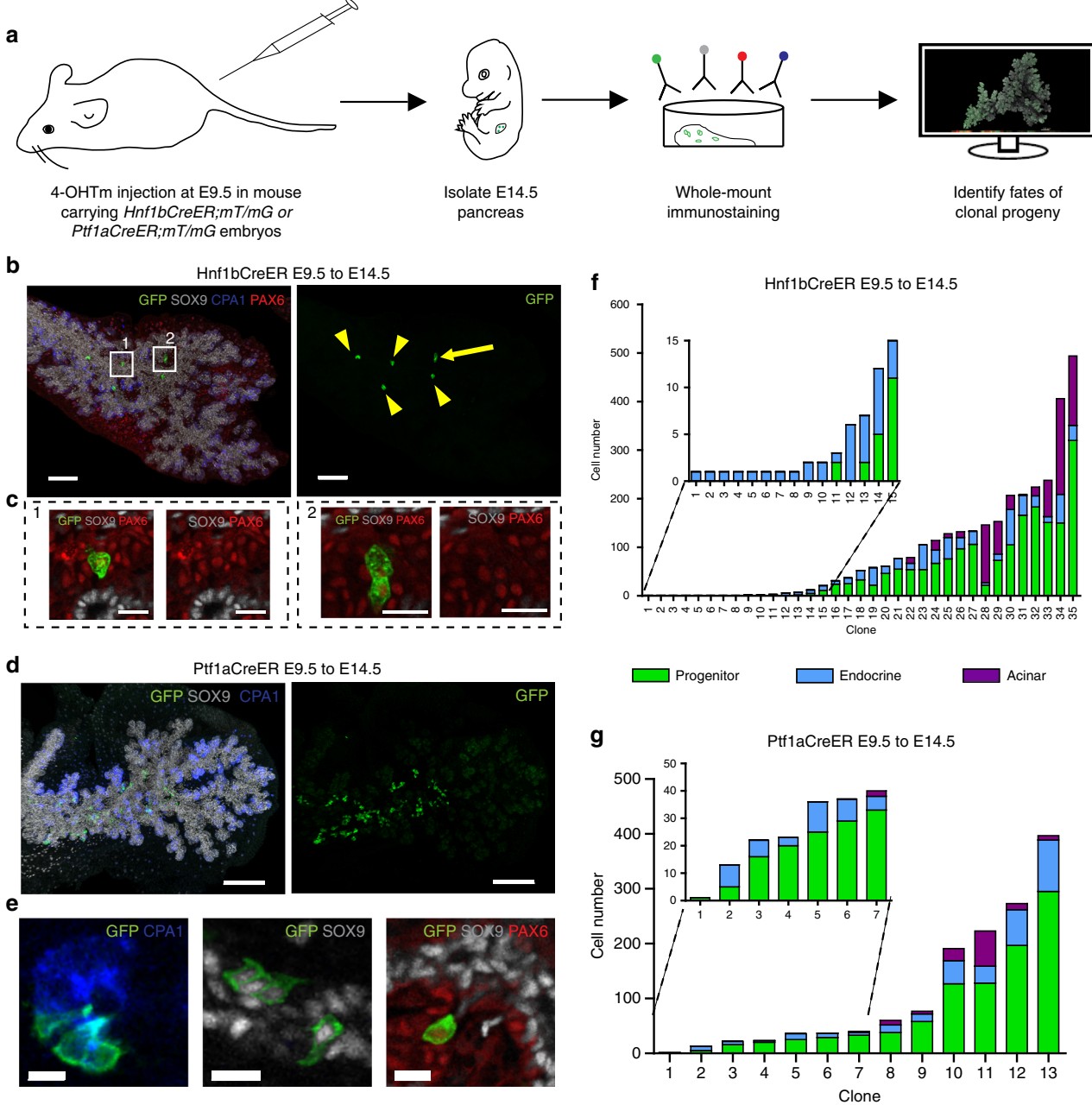

**Fig. 4** *Hnf1b*$^{CreER}$- and *Ptf1a*$^{CreER}$-mediated clonal analysis confirms the existence of ducto-endocrine bipotent clones and the absence of unipotent acinar progenitors at E9.5. **a** Schematic overview of strategy applied to identify fates of clonal progeny from E9.5 pancreatic progenitors. **b** 3D MIP and high-magnification optical sections (**c**) of clonal progeny from a unipotent endocrine clone containing six labelled endocrine PAX6$^+$ endocrine cells generated by *Hnf1b*$^{CreER}$-mediated clonal analysis. Arrow indicates two juxtaposed GFP$^+$ cells, whereas arrowheads indicate single GFP-labelled cells. Scale bars, 100 μm **b** and 15 μm **c**. **d** 3D MIP and high-magnification optical sections (**e**) of a multipotent clone derived from *Ptf1a*$^{CreER}$-mediated lineage tracing. Scale bars, 150 μm **d** and 15 μm **e**. **f**, **g** Quantification of clone sizes and fate compositions following *Hnf1b*$^{CreER}$- and *Ptf1a*$^{CreER}$-mediated clonal analysis, respectively (*n* = 120 embryos for *Hnf1b*$^{CreER}$ and 34 embryos for *Ptf1aCreER*- the images displayed show representative data from those). Note that clone no. 1 contains only one cell and is only SOX9$^+$. Most clones were found in the dorsal pancreas, except clones no. 14, 17, 18, 26, 33 in **f** and clones no. 3, 8, 9 in **g** which were found in the ventral pancreas

*Ptf1a* expression at around E9.5 do not immediately form endocrine cells, unlike progenitors traced by *Rosa26*$^{CreER}$ and *Hnf1b*$^{CreER}$, though they retain endocrine differentiation capacity as these cells give rise to endocrine-containing clones later in their clonal evolution. This hypothesis is supported by *Ptf1a* anti-correlation with early markers of endocrine differentiation such as *Mfng* and *Neurog3* in our single-cell qRT-PCR analysis at E9.5 (Supplementary Fig. 3).

**A probabilistic model of progenitor progeny fate allocation.** The apparent lack of tip-trunk biased progenitors suggested by both single-cell analysis and tracing at E9.5 led us to investigate whether a model of probabilistic cell-fate choices could recapitulate the in vivo clonal distribution data. To this end, we constructed a mathematical model of in silico clonal growth by simulating cell divisions over a period spanning the in vivo clonal tracing. Every time a cell gave rise to progeny through cell division, clonal progeny were fate-allocated with a probability of

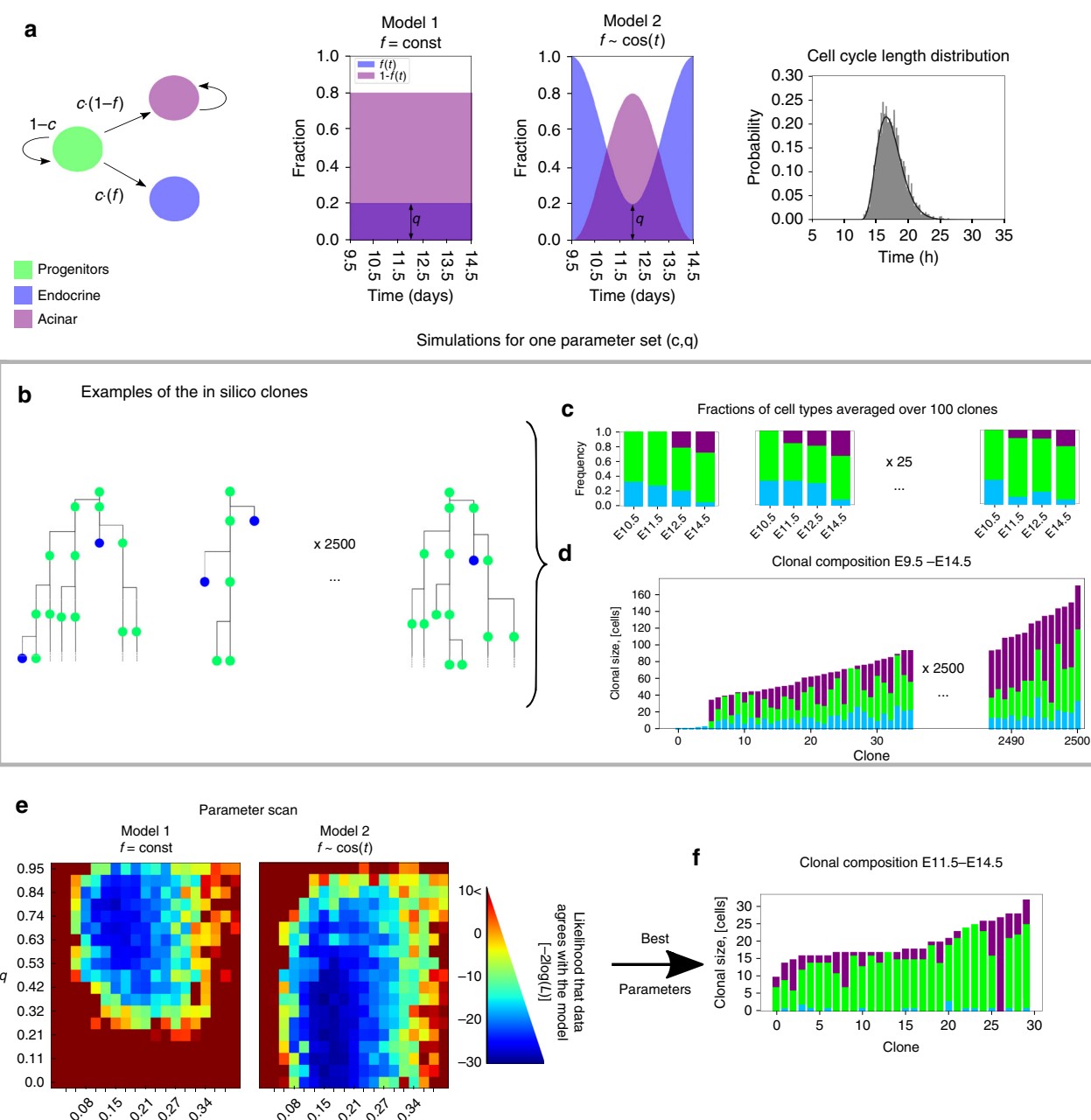

**Fig. 5** Stochastic model of clonal expansion. **a** Probabilistic transitions among three states are presented in a state diagram: at the division progenitors (*green*) differentiate with probability *c* and maintain progentor state with probability 1 − *c*. When differentiating, they become endocrine (*blue*) with probability *f* and acinar (*purple*) with probability 1 − *f*. We compare two models: in Model 1 the probability *f* is constant between E9.5 and E14.5, and in Model 2 *f* is time-dependent and has a minmum at around E12. The height of the minimum is characterised by parameter q. In Model 1 *f* = *g*. See Methods for the exact functional form of *f* for Model 2. While acinar cells continue replicating, endocrine cells are assumed not to replicate. For all replicating cells the cell-cycle lengths are drawn from the gamma distribtuion from[19] (right panel). For every parameter set both models were simulated 2500 times. **b** Examples of the in silico clonal lineages. **c** Fraction of cell types from individual clones at E9.5, 11.5, 12.5 and 14.5 (corresponds to experimental data in Supplementary Fig. 7d). **d** Clonal composition at E14.5 (corresponds to Figs. 1h and 4f). **e** For each of the parameter sets *c*, *q* we estimate the likelihood of the model fitting the data (See Methods and Supplementary Figs. 8–10). The results of the parameter scan are quantified by the log-likelihood, −2log(*L*). Parameter scans show that Model 2, where the differention of acinar cells and endocrine cells change with time, is more likely. Using Akaike Information Criteria score, AIC, we find that Model 2 is better at describing the data (AIC$_1$ = −14.5 and AIC$_2$ = −22.1). The probability that the two models are equally good is *p* = 0.02. **f** The model predicts that if clonal anlyses are started at E11.5–12 instead of E9.5, it becomes more likely to observe lineages fully commiting to the acinar fate

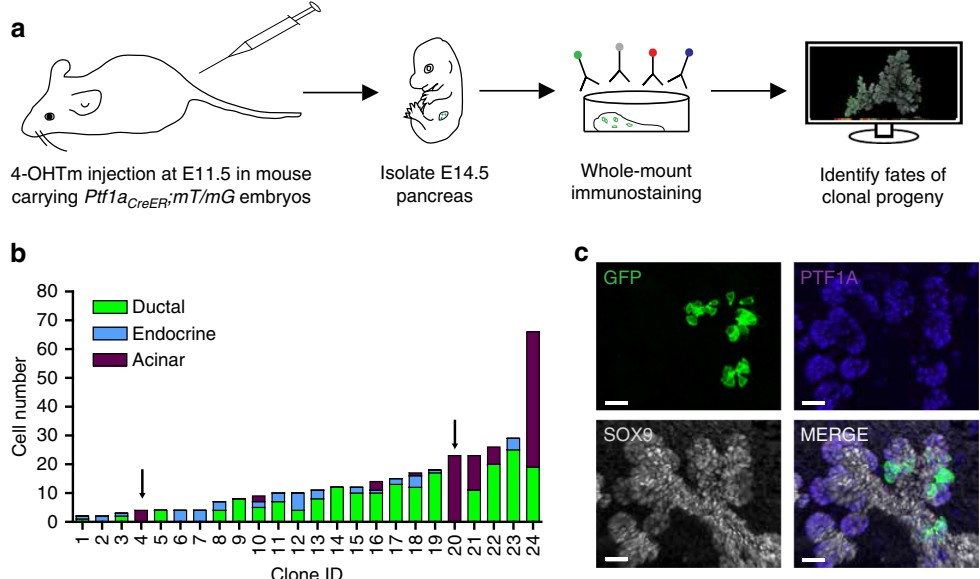

**Fig. 6** *Ptf1a*[CreER]-mediated clonal analysis identifies unipotent acinar progenitors at E11.5. **a** Schematic overview of strategy applied to identify fates of clonal progeny from E11.5 pancreatic progenitors. **b** Quantification of clone sizes and fate compositions following *Ptf1a*[CreER]-mediated clonal analysis ($n = 24$ clones in 70 embryos analysed). Arrows indicate two acinar unipotent acinar clones, as predicted by our mathematical model. **c** A representative 3D MIP of one of the two acinar unipotent clones. Most clones were found in the dorsal pancreas, except clones no. 2, 9 and 16 which were found in the ventral pancreas. Scale bars, 20 μm

differentiating *c* and a probability of becoming endocrine *cf* or acinar $c(1 - f)$ (Fig. 5a). Here *f* is a bias towards endocrine fate; *f* = 0.5 means that cells are equally likely to be allocated exocrine or endocrine fates. Simulated cell cycle lengths were randomly drawn from a gamma-distribution based on our measurements in vitro[19] and those of Bankaitis et al.[30] in vivo. Progeny committed to the endocrine lineage were approximated to be non-proliferative while acinar cells proliferated. For simplicity the acinar proliferation rate was approximated to be similar to progenitors, which is a small underestimation[15]. In total 2500 clones were simulated spanning a parameter space of probabilities for both *c* and *f* (Fig. 5b, d). Two models were compared, one with fixed probability of becoming endocrine rather than exocrine *f = q*, or one where *f* varied over time with a minimum *q* around E12 (Fig. 5a). The minimum around E11.75-E12 is suggested by the observation that the number of NEUROG3-expressing cells has a minimum at this time point[31].

To quantitatively compare simulation results with the data, we focused on two types of datasets. First, histograms in Figs. 1h and 4f contain information about the clonal variance in fractions of acinar and endocrine cells at one time point E14.5. For simplicty we focused on the variance in fractions of acinar cells and to increase sample size combined the datasets in Figs. 1h and 4f into one. Second, the staining of pancreata at four time points in Supplementary Fig. 7d does not contain information about the clonal variance but reperesents the typical cell fractions. To compare our models with the first data set we recorded the acinar fraction from simulated lineages with at least one acinar cell for each parameter set (Supplementary Fig. 8a–e). We estimated the underlying probability density function (PDF), shown on top of the histogram in Supplementary Fig. 8e by Kernel Density estimation with bandwidth 0.5 (see Supplementary Fig. 9 and methods for details).

To compare our models with the second data set, we grouped together 100 clones to approximate the data from stained pancreata in Supplementary Fig. 7d. Here we used both acinar and endocrine fractions at E10.5, 11.5, 12.5 and 14.5 to estimate PDFs as described above. This allowed us to estimate the likelihood that the experimental data points came from the PDF

derived from the simulations. In other words we estimated how likely it is for simulations to produce exactly those fractions observed in vivo. The likelihood that both datasets agree with the model was a product of each of the two likelihoods (Methods, Model Implementation). Spanning a parameter space for *c* and *q* (Supplementary Fig. 10), we observed that both the model 1 with fixed endocrine/acinar probability and the model 2 displaying temporal variations in this ratio had a parameter space of good likelihood for *c* and *q* (Fig. 5e). However, the model 2 with a time-variable probability of becoming acinar peaking at around E12 was superior at describing the data according to the Akaike Information Criteria (AIC, see Methods). The shape of the optimal parameter space is also in support of model 2: once the probability to become acinar is set to peak around E12 (model 2), the performance of the model becomes less constrained by parameter *q*. The statstical approach used allows us to identify the best model, but a combination of limited amount of biological data and high stochasticity prevents us from statistically testing how well each model match the data. Taken together, our mathematical modelling suggests that the clonal analysis data are compatible with a model of probabilistic cell fate choices and predicts that when the probability of becoming endocrine is low at around E12, the progenitors most efficiently commit to the acinar lineage at this time point.

**Acinar-committed cells are detected from E11.5 to 12.** According to the model prediction, acinar-committed cells should be undetectable at E9.5, as we have seen, but should be easily identified by clonal lineage tracing from E11.5 (Fig. 5f). Previous non-clonal tracing with *Ptf1a*[CreER] suggested that all cells expressing *Ptf1a* were acinar-committed at E14.5–E15, whereas some were still multipotent at earlier time points[15]. However, previous observations did not address whether some cells may be acinar-committed earlier. Tracing using *Ptf1a*[CreER];*mT/mG* mice injected with 4-OHTm at E11.5 revealed bipotent and tripotent clones, as at earlier time points and also showed that 8% of the PTF1A-traced cells were already acinar-committed (Fig. 6),

reinforcing the notion of heterogeneity in progenitor behaviours at the clonal level during pancreas organogenesis.

**Spatial differences in proliferation impacts clonal growth**. In addition to the compatibility of the in vivo clonal heterogeneity with a probabilistic model of cell cycle progression and cell fate allocation, we questioned whether spatial patterns of differential proliferation rates might also impact clone size. To interrogate the proliferative capacity of pancreatic epithelial subdomains we performed a label dilution experiment using *Pdx1-tTA;tetO-H2B-GFP* embryos (Fig. 7a). These embryos display ubiquitous H2B-GFP expression in *Pdx1+* progenitors, however upon doxycycline (Dox) administration, H2B-GFP expression is suppressed and

will be linearly diluted by equal partition to daughter cells upon cell division[32]. By tracing the extent of label dilution from E9.5 to E12.5 and E14.5, we observed label retention in SOX9NegE-CADLow cell clusters corresponding to non-proliferative endocrine cells derived early after suppression of H2B-GFP expression (Fig. 7b, c). Label dilution following continuous Dox administration from E9.5 was evident in SOX9+ progenitors compared to endocrine cells at E12.5, and this was even more apparent at E14.5 (Fig. 7b, c). At E14.5, cells located in the central portion of the SOX9+ epithelium still displayed retention of H2B-GFP signal, whereas SOX9+ progenitors in lateral branches and the more distal portion of the epithelium displayed label dilution. This was also apparent when administering Dox at E11.5 and tracing to E14.5 (Fig. 7d). These results suggest that pancreatic

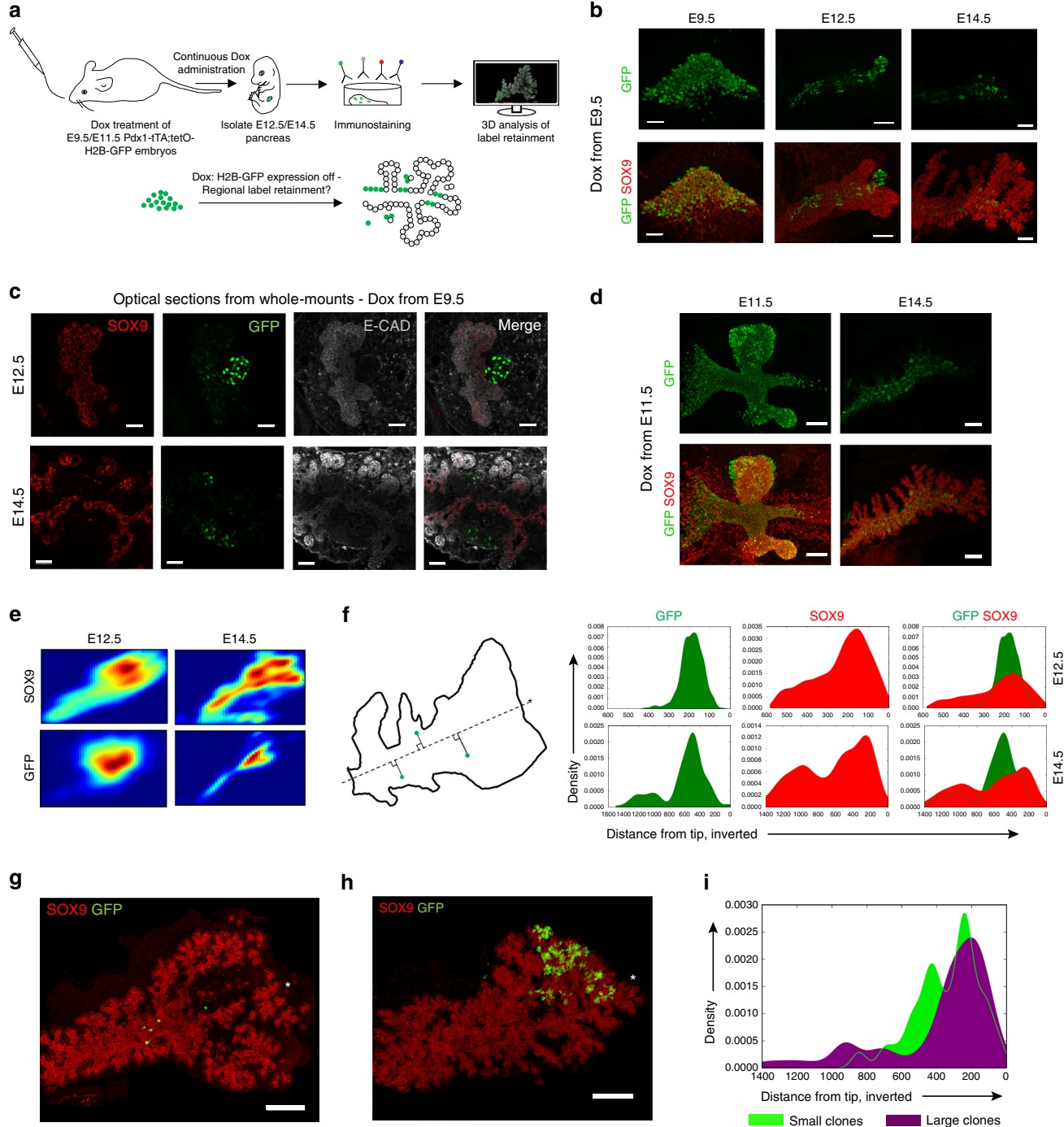

SOX9[+] ductal progenitors undergo preferential proliferation at the peripheral epithelial domains. Such preferential label retention within the central epithelial domain was additionally confirmed by plotting the 2D kernel density estimation of SOX9[+] and GFP-retaining SOX9[+] progenitors (Fig. 7e). To investigate whether the size of clones from our lineage tracing correlated with the spatial location of H2B-GFP retaining SOX9[+] progenitors, we sought to map the spatial location of clones onto the domains of differential label retention. Because of the non-stereotypic macroscopic anatomy of the pancreata between embryos, we turned to a simplistic model of spatial mapping, where we projected the location of a cell or group of cells on an axis extending from the tip of the dorsal pancreas to the duodenal root of the dorsal pancreas. This method confirms the enrichment of H2B-GFP-retaining SOX9[+] cells in a central domain of the pancreas epithelium at E14.5 (Fig. 7f) and indicates that the largest $Hnf1b^{CreER}$-derived clones tend to map to the tip (Fig. 7g–i). These results suggest that the spatial location impacts the proliferation of clonal progeny by dispersal to spatial niches with distinct proliferative capacity.

## Discussion

In this study we aimed at uncovering whether the roughly 500 cells that found the mouse pancreas contribute homogeneously to the size of the final organ and to its different functional cell types. The multipotent state of the early pancreatic progenitor population has been inferred from population-based lineage tracing studies, masking potential heterogeneity in single-progenitor contribution to organ formation[12, 15, 21]. We tested whether there are subpopulations with biases in proliferation or differentiation capacity, and whether they can be predicted by their molecular expression profile or by their initial location in the primordium.

We find that single E9.5 pancreatic cells exhibit heterogeneous contribution to organ formation, as we identify unipotent endocrine, bipotent ducto-endocrine and multipotent clones by lineage tracing at clonal density (Fig. 8). Among these categories, only the unipotent endocrine-committed cells can be predicted by single-cell molecular profiling at E9.5. These cells account for 50% of founder cells and encompass the already differentiated endocrine cells and $Neurog3$-expressing endocrine progenitors each accounting for about 12% of pancreatic cells at this stage (Supplementary Fig. 7). In addition, early endocrinogenesis encompasses other endocrine-biased cells, some of which may be replicative, possibly expressing low levels of $Neurog3$[29]. The size of this population is estimated to about 25% of all pancreatic cells based on both $Rosa26^{CreER}$ and $Hnf1b^{CreER}$ lineage tracing.

Although we did not identify any positive predictor for such endocrine-biased progenitors, $Ptf1a$ is a negative correlator based on the rarity of unipotent endocrine clones from $Ptf1a^{CreER}$-based lineage tracing, as well as the negative correlation with known endocrine specifiers from single-cell qRT-PCR (Supplementary Fig. 3). This is in agreement with the previous observation that early endocrine cells can form in the absence of $Ptf1a$[33, 34]. The fact that 50% of the cells in the emerging pancreatic primordium are biased to the endocrine lineage is surprising, since the endocrine cells make only 1–2% of the adult pancreas[26]. As the largely non-proliferative nature of endocrine-biased cells extends the time required to generate an organ of proper size, the generation of such a high fraction of endocrine cells at early stages of organogenesis contradicts expectations of optimal design theories[5]. These cells may thus carry important functions for the development of the mouse pancreas, perhaps by producing growth-stimulating components.

Despite heterogeneous and spatial differences in expression of pancreatic progenitor-associated transcription factors within the E9.5 bud, the bipotent ducto-endocrine and multipotent progenitors cannot be discriminated by single-cell qRT-PCR using our selected gene targets. Investigating more targets, protein expression or their modifications may however uncover subpopulations. Nevertheless, the heterogeneity in clone sizes and differentiation is compatible with a stochastic model of cell-fate allocation during clonal history. Comparison of several models shows that the model that best fits the data is one where cells have a probability of differentiation and where differentiation bias towards endocrine over acinar fates changes over time between E9.5 and 14.5. This would imply a double molecular gate, one controlled by the Notch pathway that controls differentiation, and a switch controlled by an unknown molecular cue that selects between endocrine and acinar fates. There is ample data supporting that Notch controls the differentiation of both acinar and endocrine lineages[10, 35–38]. In the model displaying optimal fit with our experimental data, the progenitors are predicted to have a low probability of becoming endocrine at around E11.5–E12, as supported by the progressive decrease and subsequent reappearance of NEUROG3 cells at this time point[31]. The model predicts that this corresponds to a wave of acinar cell commitment centred at around E11.5–E12 that we can experimentally capture (Fig. 6).

We also report spatial heterogeneity in progenitor proliferation which may underlie the observation of progenitors that divide only once to extreme progenies of hundreds of cells in 5 days. Recently it was demonstrated that the progeny of dividing E10.5 pancreatic progenitors in the central area of the pancreas tends to remain central but that this rule is not strict[39]. The combined

**Fig. 7** The pancreatic epithelium displays regional differential proliferation explaining impacting clone size. **a** Schematic overview of strategy implemented to identify spatial differences in proliferative capacities. E9.5 oral gavage and subsequent continuous administration of doxycycline (Dox) prevents expression of H2B-GFP in *Pdx1;− tTA/;tetO-H2B-GFP* embryos, enabling proliferation-induced label dilution in pancreatic progenitors. **b** 3D MIP of whole-mount immunostainings of dorsal pancreata at various stages following Dox administration at E9.5. Note the gradual decrease in GFP signal in SOX9[+] cells and the presence of strongly label-retaining endocrine clusters and low-retaining central progenitors, as well as the absence of label retention in the distal epithelium and in lateral branches by E14.5 ($n = 3$ at E9.5 and $n = 4$ each at E12.5 and E14.5). Representative images were extracted from those. Scale bars, 30 μm (E9.5), 80 μm (E12.5) and 150 μm (E14.5). **c** Optical sections of E12.5 (*top*) and E14.5 (*bottom*) dorsal pancreata following Dox administration at E9.5. E-CAD[Low] endocrine clusters display strong label retention, whereas label-dilution is more pronounced in the proliferative SOX9[+] progenitors. Distal lateral branches at E14.5 display complete absence of H2B-GFP retention. Scale bars, 50 μm (E12.5) and 30 μm (E14.5). **d** Following E11.5 Dox administration, the central portions of the E14.5 pancreas retains H2B-GFP signal, whereas lateral branches exhibit label dilution ($n = 3$ at E11.5 and $n = 1$ at E14.5, from which representative images were extracted). Scale bars, 70 μm (E11.5) and 150 μm (E14.5). **e** Kernel density estimation of SOX9[+] progenitors and the density of the top 10% highest GFP-retaining SOX9[+] cells. Note the central location of GFP-retaining cells. **f** One-dimensional projection of SOX9[+] progenitors and the top 10% of GFP-retaining cells onto a diagonal line running along the length axis of the dorsal pancreas demonstrate enrichment of GFP-signal in distinct domains of the pancreatic epithelium. **g**, **h** 3D MIP showing the spatial distribution of clonal progeny in a small clone in the central, proximal epithelium and a large distal clone, respectively. Scale bars, 150 μm. **i** Comparison of spatial distribution of smallest and the half of largest clones from $Hnf1b^{CreER}$-mediated E9.5-E14.5 clonal analysis ($n = 12$ clones in total)

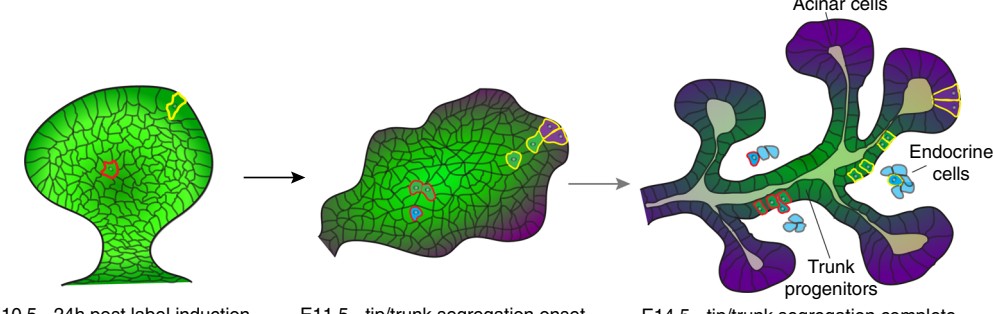

**Fig. 8** Proposed model of orchestration of pancreas organogenesis by heterogeneous clones. Although initially possessing the same intrinsic potency, the spatial position of induced clones in the pancreatic bud influences expressed in vivo potency by dispersing clonal progeny to different niches. While peripherally labelled cells (*yellow outline*) will be exposed to acinar-inducing cues concomitant with generation of trunk-fated progeny during branching morphogenesis, centrally labelled cells (*red outline*) are less likely to experience acinar-inducing signals and thus produce ducto-endocrine bipotent clones. Stochastic priming of centrally located progenitors towards the endocrine lineage might further generate heterogeneity in clonal contribution to the ductal and endocrine lineage

effect of probabilistic cell fate choices operating downstream of spatially biased progenitor proliferation and differentiation thus ultimately determines the contribution of progeny from proliferating progenitors to pancreas organogenesis.

Similar to our observations, differential potency and lineage contribution of progenitors expressing early organ markers have recently been demonstrated during heart development[40]. Our findings might facilitate the identification of niche-derived signals supporting in vitro generation of specific pancreatic cell types for regenerative medicine purposes and help elucidate the rules governing embryonic organogenesis by the concerted spatio-temporal orchestration of clones with variable contributions to organ formation.

## Methods

**Mice**. Mice (*Mus musculus*) of mixed background were housed at the University of Copenhagen. All experiments were performed according to ethical guidelines approved by the Danish Animal Experiments Inspectorate (Dyreforsøgstilsynet). The following genetically modified mouse lines were used: *Pdx1-tTA*[41], *tetO-HIST1HBJ/GFP(tetO-H2B-GFP)*[42], *Hnf1btm(CreER) (Hnf1b^{CreER})*[17], *Gt (ROSA)26Sortm4(ACTB-tdTomato,-EGFP)Luo/J (mT/mG)*[24], *Ptf1aCreERTM (Ptf1a^{CreER})*[16], *Tg(Hes1-EGFP)*[1Hri] *(Hes1-eGFP)*[43], *Gt(ROSA)26Sor^{tm1(cre/ERT2)Tyj}/J (Rosa26^{CreER})*[23]. The data were collected on male and female embryos.

**Whole-mount immunohistochemistry**. Embryonic gut tubes or isolated pancreata were fixed in 4% paraformaldehyde (PFA) for 10–30 min on ice depending on tissue size. After washing in PBS and dehydration in methanol (MeOH), fixed tissue was stored in 100% MeOH at −20 °C. Rehydrated samples were transferred to PBS + 0.5% Triton X-100 (PBST). Samples were blocked overnight at 4 °C in PBST + 1% bovine serum albumin (BSA) or in blocking solution from the Mouse-on-Mouse (MOM) detection kit (Vector laboratories) if using mouse primary antibodies (For details of antibodies and and concentrations use, please see Supplementary Table 1). Primary antibodies were incubated in PBST + 1% BSA or MOM diluent for 48 h at 4 °C. Samples were washed all day in PBST with a minimum of five washing buffer changes before addition of secondary antibodies and DNA staining dyes such as 4′,6-diamidino-2-phenylindole (DAPI) or DRAQ5. Biotinylated antibodies and secondary antibodies were supplied in PBST + 1% BSA or MOM diluent for 48 h at 4 °C followed by tissue washing and dehydration to 100% MeOH. Samples were stored at −20 °C in 100% MeOH until imaging.

**Sample clearing and imaging**. For imaging of whole-mount stained pancreata and subsequent 3D reconstruction, samples were subjected to clearing, hereby reducing light scattering. Optical clearing was performed by submerging samples in a 1:2 solution of benzyl alcohol:benzyl benzoate (BABB). Cleared samples were subsequently mounted in glass concavity slides and submerged completely in BABB to maintain refractive index matching and sample transparency. Cleared samples were imaged using a Leica SP8 confocal microscope with a 20×/0.75 oil immersion objective at 1024 × 1024 resolution. Samples were imaged in an 8-bit format unless otherwise indicated.

**In vivo clonal analysis**. 4-OH tamoxifen (4-OHTm, Sigma, H6278) was prepared as a 10 mg/mL stock solution in 10% ethanol and 90% corn oil and

subsequently diluted in vehicle solution (10% ethanol, 90% corn oil) to obtain the desired concentration. For E9.5 to E14.5 clonal analyses, mice carrying *Hnf1b^{CreER}*; *mT/mG*, *Ptf1a^{CreER};mT/mG* and *Rosa26^{CreER};mT/mG* embryos received a single intraperitoneal injection of 4-OHTm at E9.5, at a concentration of 11.5 μg/g, 57.5 μg/g and 1.35 μg/g, respectively. The dosage of 4-OHTm required to reach labelling at clonal density was initially determined by performing dose titration of E9.5 injections and analysis of clone density at E10.5 by whole-mount immunostaining. The temporal accuracy of labelling was tested using the *Ptf1a^{CreER};mT/mG*. As PTF1A expression starts at E9.5, we injected57.5 μg/g 4-OHTm at E7.5 or E8.5 and observed no labelled cell at E14.5 in 9 embryos analysed in total (4 embryos from E7.5 injection and 5 from E8.5 injection). Using Imaris™ software, GFP+ cells were identified in 3D reconstructed pancreata, and the fate of cells determined by immunostaining for various pancreatic lineage markers. For the E9.5-E10.5 short-term clonal analysis, GFP+ labelled cells were considered to be of clonal origin if one cell was seen or if the distance between recombined cells was less than 30 μm after the tracing period, based on the estimates of cell migratory capacity from Kim et al.[19]. For the mapping of clone position in the E10.5 bud, embryos harbouring one labelled cell or two labelled sister cells were considered for the analysis.

**Single-cell qRT-PCR**. E9.5 gut tube regions spanning the pancreatic bud and proximal duodenum were isolated from *Hes1-eGFP* embryos and stored in PBS on ice until all gut regions had been collected. Embryonic *mT/mG* tissue was added as a tissue spike-in to generate a bulk pellet mass preventing loss of the scarce GFP+ cell population. The pooled *Hes1-eGFP* gut tubes and *mT/mG* embryonic tissue was dissociated in 0.05% trypsin-EDTA (Gibco) containing 200 U DNase I (Roche) for 15 min at 37 °C with manual trituration using a p1000 pipette. Following dissociation, PBS + 10% FCS was added to inactivate the trypsinisation, and the single-cell suspension was centrifuged and re-suspended in PBS + 10% FCS followed by another round of centrifugation. The single-cell suspension was re-suspended in PBS + 10% FCS + DAPI to allow exclusion of DAPI+ dead cells. 260 single GFP+ cells were sorted into 96-well plates containing 5 μL CellsDirect 2× reaction mix (Invitrogen) and 0.05 U SUPERase-In™ RNase inhibitor (Thermo Fischer). 96-well plates containing single cells in CellsDirect were stored at −80 °C until ready to perform single-cell qRT-PCR reaction.

Prior to single-cell qRT-PCR, all primer pairs (Supplementary Table 2) had been validated on E14.5 bulk pancreatic cDNA using standard qRT-PCR. A mix containing forward and reverse primers for all 96 target genes were prepared in TE-buffer, generating a final concentration of 500 nM for each primer. mRNA from single cells was next subjected to one-step reverse transcription and specific target amplification according to the Fluidigm protocol 'One-Step Single-Cell Gene Expression Using EvaGreen® SuperMix on the BioMark™ HD system'. Upon loading of 96 × 96 chips, a 5-fold standard series of E14.5 bulk cDNA was added to five chip inlets, allowing identification of specific gene product detection by comparison of melt profiles of single-cell amplifications and bulk reactions. Using Fluidigm Real-Time PCR analysis software, data from three independent chip runs were combined, and individual reactions were passed (203)/failed (57) according to software peak detection and melt peak temperature being in range with bulk reactions. Expression of housekeeping genes was used as inclusion criterion for downstream analysis of individual cells. Single-cell qRT-PCR data were subsequently analysed using Fluidigm SINGuLAR™ Analysis Toolset, while global gene correlation tSNE-mediated dimensionality reduction was performed using a Python-based script (code available upon request). For the analysis of *Pdx1*+ pancreatic progenitors, cells were categorised as being positive for *Pdx1* if displaying CT values <20 for *Pdx1*.

**Model implementation**. We started simulating each clone from a cell in a progentor state and when comparing with clonal data (Figs. 1h and 4f), we only included clones with at least one progenitor. The model thus underestimates the number of clones fully commited to endocrine cells, but it does not affect our results since we only look at the acinar fractions. The algorithm follows the steps below.

First, start with cell in a progentor state. Second, draw a cell cycle length, $t_{cc}$, from a Gamma-distribution from[19]. To account for the unknown start of the cell cycle for the first cell, choose a random start between 0 and $t_{cc}$. Third, after time counter reaches $t_{cc}$ the cell divides and adopts one of three possible fates according to the diagram in Fig. 5a: Progenitor probability $1 - c$, acinar fate with probability $c$ $(1 - f)$ and endocrine fate otherwise. Fourth, assign two new cell-cycle lengths from the Gamma distribution. Fifth, repeat steps 3–4 for all cells.

For model 1: $f = q$, while for model 2: $f(q, t) = q + 0.5(1 - q)(1 + \cos(2\pi t))$.

To estimate a probabilityprobaiblity density function (PDF) for a distribution of discrete datapoints we use KDE. In effect it is a smootheninng step, where each data point is represented by a kernel (in our case gaussian with sigma = 6.5 for data sets from Supplementary Fig. 7d and sigma = 0.015 for combined data set Figs. 1h and 4f, also referred to as bandwidth)[44].

We find the likelihood, $L$, of an observation, $x_i$, being consistent with the model by evaluating the PDF at $x_i$, $\text{PDF}(x_i)$. The likelihood that all datapoints in a data set are consistent with the model is a product of their individual likelihoods. If there are two different data sets, their likelihoods are thus

$L_1 = \prod_i \text{PDF}_1(x_i); L_2 = \prod_j \text{PDF}_2(x_j)$, and the likelkihood of both datasets

agreeing with the model is $L = L_1 L_2$.

The AIC is a method for selecting among models. It does not give an absolute estimate of how well each of the models fits the data but $\text{AIC} = 2k - \ln(L)$ where $k$ is the number of variables and $L$ is the maximum likelihood, i.e., corresponding to the optimal parameter set[45]. The model with the lowest $\text{AIC}_c$ is the preferred model. The relative probability that an inferior model is as good as the preferred model can be calculated by use of the equation $p_i = \exp(\text{AIC}_{min} - \text{AIC}_i)$.

**Label retention experiments and image analysis**. Pregnant mice carrying *Pdx1-tTA;tetO-H2B-GFP* were subjected to oral gavage of 200 μL of 2 mg/mL doxycycline hyclate (Dox, Sigma), 3.5% vol/vol sucrose in H2O at E9.5 or E11.5, and subsequently this solution replaced ad libitum water supply to maintain repression of H2B-GFP expression. Following whole-mount immunostaining, cleared samples were imaged at 12-bit depth and subjected to 3D reconstruction and downstream analysis in Imaris™ (Bitplane). Progenitor cells were identified by SOX9 immunoreactivity, and the pancreatic epithelium was masked based on SOX9 staining in order to exclude label-retaining endocrine cells from further analysis. The xyz position of SOX9$^+$ progenitors was obtained using the Imaris™ spot detection algorithm on the SOX9-masked channel, additionally enabling extraction of mean GFP immunostaining intensity signal from the volume of the spot encompassing SOX9$^+$ nuclei. Kernel density estimation of SOX9$^+$ and the top 10% of GFP cells was applied to estimate the 2D distribution of these two cell populations. For the one-dimensional analysis of GFP retaining cell distribution, the distal-most point of the dorsal pancreatic epithelium and the centre of the dorsal pancreatic epithelium just proximal to the turning of the ductal structure connecting the dorsal pancreas to the ventral was used to extract the equation for the diagonal line running between these two points along the length axis of the dorsal pancreas. Using standard trigonometry, the xy-coordinates of SOX9$^+$ progenitors and the xy-coordinates of the top 10% GFP-retaining SOX9$^+$ cells were used to project these cells onto the diagonal line and ultimately to calculate the xy-coordinates of the intersection between the diagonal and projection line. Finally, the distance between the intersection point and the distal landmark was calculated, allowing kernel density estimation of SOX9$^+$ cells and the top 10% GFP-retaining SOX9$^+$ cells along using this one-dimensional length axis.

For the analysis of spatial distribution of clones according to total clone size, clones from *Hnf1b$^{CreER}$*-mediated E9.5–E14.5 lineage tracing amenable to analysis were classified as small and large so that both groups contained an equal number of clones.

**Quantification of endocrine precursor cell ratios**. Quantification of the ratio of endocrine precursors, namely, SOX9, NEUROG3, PAX6 and PTF1a, progenitors and acinar precursors obtained at E9.5, E10.5, E11.5, E12.5 and E14.5 (Supplementary Fig. 7). At E9.5, cells were manually counted. At E10.5–E12.5, cell numbers were determined using Imaris™ spot detection. For the quantification of putative acinar progenitors at E11.5 and E12.5, PTF1A$^{High}$ cells were quantified based on mean intensity of nuclear PTF1A above 80 *grey* scale values from 8-bit format images. This pixel intensity threshold was selected based on the intensity of PTF1A$^+$ cells segregated to the periphery at E11.5 and E12.5, although PTF1A$^+$ displaying mean intensity values above the threshold are still found scattered within the central epithelium. At E14.5, absolute cell numbers were determined using a custom built image segmentation and analysis software.

**Neighbour identification by Voronoi-Delaunay triangulation**. The xyz coordinates of E9.5 pancreatic progenitors were obtained after manual spot detection of SOX9$^+$ nuclei in 3D reconstructed images of E9.5 gut tubes. The

mean fluorescence intensity of the applied staining for pancreatic transcription factors were extracted from the spot volume. Voronoi-Delaunay triangulation was implemented using a python-based script, and a 10 μm distance threshold was applied in order to identify nearest neighbours with biological meaning. The coefficient of variation, as well as the mean intensity of neighbours was computed from corresponding fluorescence intensities of neighbour-connected cells, in order to visualise spatial patterns of heterogeneity and regionalisation of transcription factor expression levels.

**Code availability**. Python and Python-Notebook code files, along with an explanatory Read-me file, linked to Fig. 5 are provided as Supplementary Software, under the GNU General Public Licence (GPL). Codes are available upon request for label retention experiment quantifications and Voronoi-Delaunay triangulations.

**Data availability**. The authors declare that all data supporting the findings of this study are available within the article and its Supplementary Information files or from the corresponding author upon reasonable request.

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

## Acknowledgements

We thank Pau Rué, University of Cambridge, UK, for his initial guidance on modelling, Alfonso Martinez Arias, University of Cambridge, UK, for enabling the mathematical modelling training, Jutta Bulkescher for imaging guidance and Gopal Karemore for automated image segmentation. This project was supported by the Novo Nordisk Foundation and grant 12–126875 from Det Frie Forskningsråd-Sundhed og Sygdom to A.G.B. and the Danish National Research Foundation/Danmarks Grundforskningsfond, grant DNRF 116 to A.T. and A.G.B.

## Author contributions

H.L.L., A.G.-B. and Y.H.K. designed the project. H.L.L. performed and analysed most experiments except those specified below. L.M.-C. performed an analysed the experiments leading to Fig. 1. A.V.N, guided by A.T. performed the *in silico* modelling leading to Fig. 5 and Supplementary Figs. 8–10. Y.H.K. guided experiments and contributed to single-cell qRT-PCR. C.V.W. provided the *PTF1a^CreER* mice. H.L.L. and A.G.-B. wrote the manuscript which was commented and approved by L.M.-C., Y.H.K., A.V.N., A.T. and C.V.W.

## Additional information

**Competing interests:** The authors declare no competing financial interests.

