## [Peer Review File · Nature Communications]

Reviewers' Comments:

Reviewer #1 (Remarks to the Author)

In this submission the authors characterize putative endocrine progenitors present at E9.5 in the dorsal pancreatic bud by determining their clonal fate and size and by molecular profiling using single cell transcriptomics. They report the identification of three types of pancreatic progenitors that differ in their lineage potential and clone size. Interestingly, they report the co-existence of bipotent ducto/endocrine and multipotent progenitors, which are indistinguishable at the transcriptional level with their chosen marker set at E9.5. In contrast, unipotent, endocrine committed cells are molecularly distinct and make up half of the cells present in the dorsal bud at E9.5. In addition, the authors show that spatial location/excess to niche signals impacts proliferation (clone size) as well as lineage commitment to acinar cells. Together this is a very interesting and important study, which provides mechanistic insight into progenitor expansion and endocrine lineage specification.

Major comments:

- It is well known that endocrine lineage formation happens during first and second transition in the pancreas. Moreover, the dorsal and ventral buds are specified via different neighbouring tissue interactions. As this study covers different phases of endocrine induction and also different tissues (dorsal bud) the necessary information should be mentioned in the introduction. The authors should discuss their results in the context of lineage specification in a spatio-temporal model, i.e. alpha cells during primary transition are not thought to contribute to islets. Would the analysis of the ventral bud progenitors give the same results? What would be the potency and heterogeneity during secondary transition? Are the authors proposing a general or specific model of endocrine lineage specification?
- As the authors mention, it is very surprising that half of the cells in the pancreatic anlage are endocrine committed cells, although endocrine cells make up only 1-2% of the adult pancreas. Is this due to the labelling of the cells during primary transition? What would one expect when one labels the cells at the beginning of secondary transition? Do the authors they would get the same results? Please discuss.
- I am wondering if there is any (functional) difference between endocrine cells generated from multipotent/bipotent progenitors and endocrine cells derived from unipotent, endocrine committed progenitors. Are they long-lived? Do they contribute to mature islets? Do they allocate to a specific endocrine lineage?
- From the sc qPCR one would expect that mainly Gcg+ cells are formed during primary transition. There are 4 Gcg+ but also 2x Ins+ cells identified. How do the authors think that the unipotent endocrine fate is determined? Temporal and/or spatial regulation.
- It is also surprising that only a few endocrine committed progenitors are detected in the single cell qPCR, this is contradictory to the lineage tracing data. What is the explanation for this discrepancy.

- If the authors want to generalize on endocrine specification in the pancreas the clone behaviour during secondary transition needs to be analyzed by labelling at E12.5. If not please tune down the conclusions.

Minor comments:

- I would suggest to include in Figure 1 a pancreatic and endocrine lineage allocation model during primary and secondary transition as it is very difficult to follow if one is not familiar with all the transcription factors.
- Figure 1a is not mentioned in the text.
- Use small letters for Figures in the text as it is in the Figures.
- Is Pax6 expressed in all endocrine cells at E14.5? Pax6 staining does not look convincing.
- I am confused about the term progenitor in Figure 1e. Do you mean Sox9+ duct cells by progenitor?
- Ngn3+ cells and the endocrine lineage is not post-mitotic, maybe they slow down in cell cycle, but how would it be possible that even after birth there is still a high percentage of proliferation in the islets.

Reviewer #3 (Remarks to the Author)

This manuscript describes a clonal analysis of the pancreatic progenitors that have just been specified at E9.5. The authors find that these progenitors display heterogeneous behaviours, with some cells only giving rise to endocrine descendants that rarely divide, while other progenitors can contribute acinar, endocrine and ductal fates. They then go on to propose that this heterogeneity arises at least in part due to a stochastic response to patterning cues, as those cells that acquire an acinar fate seem to be biased by their location.

I have two concerns with this paper. First, that although this is a very carefully performed study, it is quite descriptive and in my view does not add substantially to what is already known, such as the very elegant work the authors published in PLoS Biology last year (Kim et al, 2015). Also, the evidence for the conclusion that cells at the epithelial periphery are biased to become acinar is not overwhelming. This assumption could be strengthened by functional studies.

Second, and less importantly, the labelling technique used offers the potential to mark cells well after the time of injection, that could confound some of the conclusions of the manuscript. The reported perdurance of tamoxifen activity well after the time of injection (Reinert et al., PloS One 2012) could lead to labelling of cells after E9.5. This post-injection labelling could contribute to a proportion of the single labelled cells.

Reviewer #4 (Remarks to the Author)

The manuscript describes cell proliferation and differentiation in embryonic mice pancreas primarily based on expression data and labelling experiments, alongside a computational model that the authors feel describes their dataset well. The major conclusion from the manuscript is that a relatively simple "single progenitor" model describes the data well when combined with spatial signalling information. These local signals drive different proliferation rates and paths of differentiation. The paper is interesting and exciting, and the model for action plausible, but I have some concerns about the relationship between the computational model and the data that need to be addressed in future revisions.

The experimental data appear sound, and differential gene expression and proliferation rates are shown using different techniques. The authors note that not all cell types can be distinguished using the marker set and suggest that this indicates that bipotent and multipotent cells may not be molecularly distinct. This is not my area of expertise, but from what I understand I am largely comfortable with the results as presented. I feel however that the statements regarding the uniqueness of different cell populations may be overstated and should be softened; whilst expression of markers may be broadly similar, the cells may be molecularly distinct through other elements of cell state (e.g. phosphorylation state of proteins, translation, etc). I also note that some figures referenced in the manuscript are missing (e.g. supplementary fig 8)- I suspect that this should be SF7 but it should be corrected.

The computational model and the links between this model and the data are substantially less clear. The authors do not report their models in sufficient detail to be reproduced. In particular, they note only a single quantity (division rate), and not other quantitative parameters of the model. The model and its parameters should be described clearly and formally somewhere in the text. Relatedly, the authors note in the checklist that code availability is described in the text; however each piece of software used in the study is only available on request. At a minimum, a binary with input files should be made on a public site to allow others to replicate the work. If the model parameters are hardcoded, the code itself would need to be made available. The source of the quantitative parameters (rates, probabilities etc) should also be noted clearly, particularly if their are fitted parameters.

Furthermore, it is not clear what, if any, quantitative agreement is tested between the data and the model. If the model is only intended as a qualitative description that broadly matches the experimental data this needs to be stated explicitly as the abstract implies otherwise. Quantitative comparisons could include recapitulating the distribution or average of clone sizes observed at different time points.

Reviewer #1 (Remarks to the Author):

In this submission the authors characterize putative endocrine progenitors present at E9.5 in the dorsal pancreatic bud by determining their clonal fate and size and by molecular profiling using single cell transcriptomics. They report the identification of three types of pancreatic progenitors that differ in their lineage potential and clone size. Interestingly, they report the co-existence of bipotent ducto/endocrine and multipotent progenitors, which are indistinguishable at the transcriptional level with their chosen marker set at E9.5. In contrast, unipotent, endocrine committed cells are molecularly distinct and make up half of the cells present in the dorsal bud at E9.5. In addition, the authors show that spatial location/excess to niche signals impacts proliferation (clone size) as well as lineage commitment to acinar cells. Together this is a very interesting and important study, which provides mechanistic insight into progenitor expansion and endocrine lineage specification.

Major comments:

- It is well known that endocrine lineage formation happens during first and second transition in the pancreas. Moreover, the dorsal and ventral buds are specified via different neighbouring tissue interactions. As this study covers different phases of endocrine induction and also different tissues (dorsal bud) the necessary information should be mentioned in the introduction.

We have revised the introduction at the bottom of page 2 to introduce the fact that the dorsal and ventral buds are induced by different mechanisms, without getting into details because it is not the specific focus of the article. We currently mention endocrine cell induction both in the early bud and in bipotent progenitors. We would rather not use the terms primary and secondary transition. This terminology referred essentially to the two waves of endocrine cell production but the literature has evolved a lot and many other events including the polarization/formation of lumen, presence of multipotent versus bipotent progenitors at the two stages were not part of the primary/secondary transition definition. We prefer to refer to developmental time and explain the events occurring at specific time points.

The authors should discuss their results in the context of lineage specification in a spatio-temporal model, i.e. alpha cells during primary transition are not thought to contribute to islets. Would the analysis of the ventral bud progenitors give the same results? What would be the potency and heterogeneity during secondary transition? Are the authors proposing a general or specific model of endocrine lineage specification?

With regards to the endocrine cells formed at the primary transition, their numbers are smaller than the number of cells formed at the secondary transition. Nevertheless, it is not so clear that they do not contribute to islets. For example, lineage tracing of cells labelled by Neurog3-CreER participate in adult islets whether they are marked at the primary or secondary transition (Gu et al., Development 2002).

With regards to the ventral bud, we found both bipotent and tripotent clones from both the dorsal and ventral bud. However, no endocrine-committed clone was found ventrally by tracing from E9.5, likely

marking a later endocrine differentiation onset ventrally. This was added to the manuscript at the bottom of page 5 and specific experimental information regarding ventral clones was added in figure legends.

At the secondary transition, it was previously shown by global lineage tracing using Sox9-CreER and Hnf1b-CreER that there are bipotent progenitors generating endocrine and progenitor/ductal cells. This is also suggested by our 2015 study, though not long term traced. These cells do not contribute to acinar cells any more from E14.5. In contrast Ptf1a-CreER lineage tracing shows that Ptf1a-expressing cells do not contribute endocrine and ductal cells any longer at this stage and are restricted to the acinar fate. We have investigated this transition further here at the intermediate stage at E11.5, addressing whether the loss of acinar competence of Ptf1a-expressing cells is lost abruptly or progressive. Our results show that the transition is progressive and that there are already some Ptf1a-expressing cells that are acinar-committed at E11.5. These are shown in the new figure 6.

We have revised our in silico model (see below, response to reviewer 4). Our model is still a stochastic model in which endocrine cells and acinar cells differentiate from progenitors with a certain probability, which reflects the likelihood that they encounter specific signals. However, we now show that the model that best fits is one where only endocrine cells differentiate both at the primary and secondary transition, and acinar cells enter the differentiation path between E11.5 (data shown here) and E14.5 (Zhou et al. 2007; Pan et al. 2013; Solar et al., 2009; Kopinke et al., 2011; Kopp et al. 2011). Previous work has shown that no new progenitor enters the acinar differentiation path after E14.5.

- As the authors mention, it is very surprising that half of the cells in the pancreatic anlage are endocrine committed cells, although endocrine cells make up only 1-2% of the adult pancreas. Is this due to the labelling of the cells during primary transition? What would one expect when one labels the cells at the beginning of secondary transition? Do the authors they would get the same results? Please discuss.

The early pancreatic bud in the primary transition is indeed special. We have not performed tracing with Rosa26-CreER at the secondary transition but our published tracing at E13.5 shows that 25% of Hnf1b-CreER traced cells contribute endocrine cells within 24h (NEUROG3). The percentage is thus relatively high though underestimated by the fact that cells are followed for only 24h. At E9.5, this paper shows in supplementary figure 6b that 23% of Hnf1b-CreER traced cells (counting one and two cell clones, 14 endocrine, 46 progenitors) contribute endocrine cells within 24h (NEUROG3). The endocrine conversion rate is thus similar at these two stages. However, it is expected to decrease to almost 0 between the primary and secondary transition at E11.5 since the Cleaver lab has shown a transient decrease and absence of Neurog3 at this stage (Villasenor, Dev. Dyn. 2008). This corresponds to the time of acinar cell emergence. Our in silico model has taken this change in endocrine and acinar differentiation ability into consideration, and led us to revise the discussion.

- I am wondering if there is any (functional) difference between endocrine cells generated from multipotent/bipotent progenitors and endocrine cells derived from unipotent, endocrine committed

progenitors. Are they long-lived? Do they contribute to mature islets? Do they allocate to a specific endocrine lineage?

Although our study does not directly investigate this, we think this is likely. The clones that form only endocrine cells differentiate right away and our previous work (Johansson, Dev. Cell 2007) shows that they make mostly alpha cells. This work also shows they remain at least until E18.5. The early unipotent clones are thus likely to produce largely alpha cells. Lineage by Gu and Melton (Gu et al., Development 2002) shows that even the early endocrine progenitors expressing NEUROG3 at E8.5 can contribute to 8 week-old adult islets. We have not analyzed whether they give rise to a specific lineage which would lead us to do as many experiments as we conducted here and would dilute the focus of the article.

- From the sc qPCR one would expect that mainly Gcg+ cells are formed during primary transition. There are 4 Gcg+ but also 2x Ins+ cells identified. How do the authors think that the unipotent endocrine fate is determined? Temporal and/or spatial regulation.

Indeed, as studied in our previous paper (Johansson, Dev. Cell 2007), and in other articles, the endocrine cells formed at the primary transition are mainly GDG+. GCG and INS protein are found together in 30% of these cells, which has been seen by us and many others. Recent single cell sequencing suggests that some level of hormone co-expressing states remain even in the early postnatal period (Wang/Kaestner, Cell Metabolism 2016). Our present work does not address how unipotent fate is determined.

- It is also surprising that only a few endocrine committed progenitors are detected in the single cell qPCR, this is contradictory to the lineage tracing data. What is the explanation for this discrepancy.

This analysis was more aimed at finding subpopulations among progenitors. To isolate the pancreatic bud from other tissues and collect enough cells, we had to resort to FACS-sorting and we used a HES1-GFP mouse line to isolate the GFP+ epithelial cells. This reporter gene is progressively silenced as Notch signalling is abrogated during endocrine differentiation, and thus most endocrine cells are excluded from the analysis due to the loss of GFP fluorescence.

- If the authors want to generalize on endocrine specification in the pancreas the clone behaviour during secondary transition needs to be analyzed by labelling at E12.5. If not please tune down the conclusions.

The first signs of secondary transition are initiated at E11.5, when NEUROG3 expression is transiently silenced and re-starts by E11.75. To capture this phase we performed tracing at E11.5. We observe that the same type of clones are detected, including bi- and tri-potent clones. However, these experiments also showed that some cells are already acinar-committed by that stage. This enabled us to revise our in silico model to take this into consideration. See comment above and new figure 5.

Minor comments:

- I would suggest to include in Figure 1 a pancreatic and endocrine lineage allocation model during primary and secondary transition as it is very difficult to follow if one is not familiar with all the transcription factors.

We added this model that will indeed help the readers, especially non-specialists.

- Figure 1a is not mentioned in the text.

We referenced this panel in the text.

- Use small letters for Figures in the text as it is in the Figures.

This has been corrected.

- Is Pax6 expressed in all endocrine cells at E14.5? Pax6 staining does not look convincing.

In general, nuclear PAX6 was clear (see Figure 1b', many panels in figure 4 and supplementary figure 7e and f) but we agree that there is some background in figure 1c'). We could not find a better illustration and it is much easier to see on 3D images that can be rotated, which is the data we used for quantification. As indicated in the text, we used a combination of PAX6, absence of/low SOX9 and being at the periphery of the epithelium to qualify endocrine cells. We are thus confident in the tracing though we would have preferred to show a better image.

- I am confused about the term progenitor in Figure 1e. Do you mean Sox9+ duct cells by progenitor?

Yes, we call progenitors the SOX9-positive cells. We changed the text on page 3 and the figure legend on page 12 into "progenitors constituting the SOX9+ ducts" to better match the figure.

- Ngn3+ cells and the endocrine lineage is not post-mitotic, maybe they slow down in cell cycle, but how would it be possible that even after birth there is still a high percentage of proliferation in the islets.

The reviewer is absolutely right. We edited the text either removing the term post-mitotic (on pages 2 and 3). On page 4, 5, and 7, we replaced it by non-proliferative, which is a statement that matches the observations that no cell division occurred but does not preclude that they may divide later.

--

Reviewer #3 (Remarks to the Author):

This manuscript describes a clonal analysis of the pancreatic progenitors that have just been specified at E9.5. The authors find that these progenitors display heterogeneous behaviours, with some cells only giving rise to endocrine descendants that rarely divide, while other progenitors can contribute acinar, endocrine and ductal fates. They then go on to propose that this heterogeneity arises at least in part due to a stochastic response to patterning cues, as those cells that acquire an acinar fate seem to be biased by their location.

I have two concerns with this paper. First, that although this is a very carefully performed study, it is quite descriptive and in my view does not add substantially to what is already known, such as the very elegant work the authors published in PLoS Biology last year (Kim et al, 2015).

We would like to provide some clarifications regarding the important differences between the two studies. The 2015 study did not address single-cell contributions at the start of organ formation (a question never addressed in any other organ), did not deal with the question of multipotent progenitors and thus acinar cell birth (the main finding of the previous paper was that asymmetric cell division is an epiphenomenon of stochastic endocrine birth) and was not tracing multiple generations of progeny but only the immediate daughters. There was also no single cell PCR addressing the issue of predetermination. Also, while reviewer #3 would have liked a more mechanistic study, the first one finds it mechanistic. There is clearly no gene knock-out, which is sometimes what is expected of a mechanistic study but we think it is not the only way to make an investigation mechanistic. Our study clarifies the mechanisms of pancreas formation: while before this work we thought the pancreatic primordium is made of multipotent progenitors expected to be molecularly the same, our experiments show that the mechanisms of pancreas formation rely on the early determination of molecularly different progenitors to the endocrine lineage, a novel and surprising proliferation capacity of endocrine-committed progenitors, and heterogeneous spatially controlled proliferation. We also added more data tracing at E11.5 showing that the tripotent cells seen at E9.5 are cells in which an acinar fate will be induced starting at E11.5 and at this stage we find clones that are strictly acinar committed. This adds additional depth to our study. In this respect, we believe that the current work adds substantial new knowledge to the field of pancreas developmental biology, and that our findings are of general interest in determining mechanisms of cell fate specification from the onset of organogenesis. The co-existence of progenitors that are multipotent *per se*, relying on spatially distributed molecular cues for downstream progeny cell fate induction, along with the presence of distinct differentiation-prone progenitors harbouring proliferative capacity, might serve as a general principle for optimized organogenesis by allowing terminal differentiation concomitant with progenitor expansion and organ primordium growth. As the current literature on mechanisms of cell fate specification during vertebrate organogenesis are very limited, our work holds important implications for future interrogation of other organ systems and the molecular cues guiding spatially controlled cell differentiation.

Also, the evidence for the conclusion that cells at the epithelial periphery are biased to become acinar is not overwhelming. This assumption could be strengthened by functional studies.

We do apologize for the wording, which was misleading. Indeed, we do not see a bias indicating that a clone located at the periphery of the pancreas at E9.5 would become acinar. We actually see quite an important dispersion of clones. What we meant was that since PTF1a becomes enriched at the periphery after E11.5 (former figure 5c) and even more at E12.5, if a cell of a clone migrated to the periphery it may increase its chances of becoming PTF1a-high and exocrine. However, we have analyzed the position of clones with acinar cells from the new tracing at E11.5 and we do not see any clear bias. In the revision period, we also conducted live imaging from E9.5 for about 48h and we see a lot of cells moving from the superficial to more internal layers and vice versa. We thus removed all references to peripheral location biasing the cells to the acinar lineage. However, based on the label retention experiment, we maintain that proliferation is generally greater in the most peripheral SOX9 progenitors (as seen in acinar cells, which are peripheral, by Zhou and Melton, Dev. Cell 2008).

Second, and less importantly, the labelling technique used offers the potential to mark cells well after the time of injection, that could confound some of the conclusions of the manuscript. The reported perdurance of tamoxifen activity well after the time of injection (Reinert et al., PloS One 2012) could lead to labelling of cells after E9.5. This post-injection labelling could contribute to a proportion of the single labelled cells.

We have performed experiments addressing Tamoxifen perdurance and thus accuracy of labelling time. In our experience, perdurance is not a problem with when using the active metabolite 4OH tamoxifen for injections at such a low concentration. In Reinert et al. the doses where perdurance was seen were 3x8mg tam and they show a 10-fold reduction in the perdurance already at 3x1mg. We use more than 10 times less than this lower dose! Furthermore, injecting the active 4OH metabolite should limit perdurance as the activity of the drug is limited to a narrower temporal pulse. To address this issue, we injected tamoxifen at the dose of 57.5 µg 4OH tamoxifen in the Ptf1a-CreER line. This line was used as proof of principle and since PTF1a expression starts in the pancreas at E9.5, injection of 4OH Tam at E7.5 and E8.5 was performed. Any 4OH tam still available at sufficient doses for labelling after 48 and 24h respectively should label cells at PTF1a onset at E9.5. We did not see any labelling in these conditions. We added these tests in the methods section on page 9.

Reviewer #4 (Remarks to the Author):

The manuscript describes cell proliferation and differentiation in embryonic mice pancreas primarily based on expression data and labelling experiments, alongside a computational model that the authors feel describes their dataset well. The major conclusion from the manuscript is that a relatively simple "single progenitor" model describes the data well when combined with spatial signalling information. These local signals drive different proliferation rates and paths of differentiation. The paper is interesting

and exciting, and the model for action plausible, but I have some concerns about the relationship between the computational model and the data that need to be addressed in future revisions.

The experimental data appear sound, and differential gene expression and proliferation rates are shown using different techniques. The authors note that not all cell types can be distinguished using the marker set and suggest that this indicates that bipotent and multipotent cells may not be molecularly distinct. This is not my area of expertise, but from what I understand I am largely comfortable with the results as presented. I feel however that the statements regarding the uniqueness of different cell populations may be overstated and should be softened; whilst expression of markers may be broadly similar, the cells may be molecularly distinct through other elements of cell state (e.g. phosphorylation state of proteins, translation, etc).

We softened the interpretation on page 8. Yes, we are looking at a subset of markers only and only transcriptionally. While they are sufficient to identify progenitors biased against the endocrine fate (unipotent) more unique populations may be identified by investigating more markers or using other techniques.

I also note that some figures referenced in the manuscript are missing (e.g. supplementary fig 8)- I suspect that this should be SF7 but it should be corrected.

We modified the reference. We also referred to Figure 1a (now 1b) in the text.

The computational model and the links between this model and the data are substantially less clear. The authors do not report their models in sufficient detail to be reproduced. In particular, they note only a single quantity (division rate), and not other quantitative parameters of the model. The model and its parameters should be described clearly and formally somewhere in the text. Relatedly, the authors note in the checklist that code availability is described in the text; however each piece of software used in the study is only available on request. At a minimum, a binary with input files should be made on a public site to allow others to replicate the work. If the model parameters are hardcoded, the code itself would need to be made available. The source of the quantitative parameters (rates, probabilities etc) should also be noted clearly, particularly if their are fitted parameters.

In collaboration with Ala Trusina and Alexander Nielsen, two computational biologists, we strengthened the model. The spirit of the model is still testing whether a probabilistic model of differentiation into endocrine and acinar cells can account for our observations. When new cells are born, it includes a probability of differentiating versus remaining a progenitor and if differentiation is chosen a cell has a probability of becoming acinar or endocrine. We tested a large parameter space of probabilities and scenarios where they remain the same in time or vary. The best fit to data, as tested statistically, is a model with a transient decrease of the probability of becoming endocrine with commensurate increase of the probability of becoming acinar. While previous experimental data (Villasenor et al., 2008) already suggested a transient decrease of endocrine cell differentiation at E11.5, we now show using new lineage tracing at E11.5 that the wave of endocrine commitment is contemporary. We also tried to clarify the parameters used and the biological input quantities. This lead us to completely change figure

5, add supplementary figures S8-10, change the results on page 7 and the discussion on page 9. To enable both the evaluation of our work and its replication by others, we provide two Python code files as supplemental material.

Furthermore, it is not clear what, if any, quantitative agreement is tested between the data and the model. If the model is only intended as a qualitative description that broadly matches the experimental data this needs to be stated explicitly as the abstract implies otherwise. Quantitative comparisons could include recapitulating the distribution or average of clone sizes observed at different time points.

We think the revised modelling makes better use of the model and made quantitative comparisons that test how well the model recapitulates the observations, the parameter space where it does and also compare two alternative models. Taken together, our results show that a model where the probability of becoming endocrine or exocrine change over time better fits the experimental data.

Reviewers' Comments:

Reviewer #1:

Remarks to the Author:

All my concerns have been addressed in the revised version of the manuscript.

Reviewer #3:

None

Reviewer #4:

Remarks to the Author:

The authors have improved the computational aspects of the paper substantially. As before, my review is focused on this area rather than the biological implications or experimental techniques that are outside of my expertise.

The model is a lot more clearly described, and the relationships with the data better elucidated. Briefly, the computational aspects now show that a simple single progenitor model where cells either divide to give progenitors ($p=1-c$), endocrine ($p=cf$) or acinar ($p=1-cf$). They then compare two alternative methods for calculating f that can be characterized by a parameter q , and calculated the log likelihoods of different combinations of c and q . The distribution of likelihoods suggests that both the models are more able to discriminate between the probability of self renewal than " q ", and though model 1 has a lower likelihood it has a stronger preference for a specific value of " q ". Given the simplicity of the function of f in model 2 I am hesitant to suggest that this is overfitting, but I think that a brief discussion on the varying performance of the models and the likelihood fits could be valuable, as could the explicit reporting and calculation of the parameter sets with the highest likelihoods and their respective 95 % confidence intervals.

A plot of the experimental distributions of clone size/make up against the "best" parameters for each model would make the model agreement more clear too; though the likelihood calculations for the two models will allow the best parameters for each model to be calculated, it is not in and of itself a measure that the models recapitulate the data well. There are graphs in SFig 8 that appear to show this for different populations/times, but the figure and legend are not clear enough to know what's going on. For example, I cannot see the simulation points in the graph, and I'm not sure why the PDF is calculated from the experiment and compared with simulation rather than the other way around. Intuitively, the PDF calculated from the simulation data should be better sampled and then the experimental observations noted. Also, it seems surprising that the PDFs have a "doughnut" shape; is the center of the distribution really as unlikely as points far from the edge of the distribution?

The authors argue that a time dependent value of f describes the data better than a constant value of f , and make some specific predictions (Fig 5f), though its not clear how this could be validated. I note that 5f isn't referenced in the text, and would recommend to the authors that the prediction is either integrated with the rest of the paper and experimental validation performed or suggested, or the subfigure removed if its not a testable prediction.

A minor point on the model annotation; the choice of "q" is unfortunate as the labels on the parameter scan are easily misread as "b".

The authors also submitted some code with the recent revision. This is a big improvement from before, but the code is not standard python. From reading through the files it appears to be in a python-notebook type format. The authors need to state how a user can interact with this. To be clear; I believe that the work is reproducible, and that a sufficiently dedicated and experienced researcher could get this working, but the authors need to document it better.

Finally, there are some presentation issues. A lot of the new text has typos e.g. Fig5A, last panel "lenght", page 7 "allowsallowed", page 7 "estimateestimated" page 12 "To estimate a probabilityprobabilty" etc. These should be fixed; it distracts from what is otherwise a nice, clear description of whats going on.

Furthermore, graphs need keys on the graph themselves if they are to be readable.

Reviewer #4 (Remarks to the Author):

The authors have improved the computational aspects of the paper substantially. As before, my review is focused on this area rather than the biological implications or experimental techniques that are outside of my expertise.

The model is a lot more clearly described, and the relationships with the data better elucidated. Briefly, the computational aspects now show that a simple single progenitor model where cells either divide to give progenitors ($p=1-c$), endocrine ($p=cf$) or acinar ($p=1-cf$). They then compare two alternative methods for calculating f that can be characterized by a parameter q , and calculated the log likelihoods of different combinations of c and q .

1. *The distribution of likelihoods suggests that both the models are more able to discriminate between the probability of self renewal than "q", and though model 1 has a lower likelihood it has a stronger preference for a specific value of "q". Given the simplicity of the function of f in model 2 I am hesitant to suggest that this is overfitting, but I think that a brief discussion on the varying performance of the models and the likelihood fits could be valuable,...*
 - a. We thank the reviewer for pointing this out. We agree that the observed difference in the shape of the optimal parameters is not a result of the overfitting. It is rather a result of a constraint of time-varying probability of becoming acinar, in particular data-driven requirement that it should peak around E12. Once this is satisfied, the parameter q becomes less constrained. To address this point, we have now added the following sentence to our model section: "The shape of the optimal parameter space is also in support of model 2: once the probability to become acinar is set to peak around E12 (model 2), the performance of the model becomes less constrained by parameter q "
2. *... as could the explicit reporting and calculation of the parameter sets with the highest likelihoods and their respective 95 % confidence intervals.*
 - a. To estimate 95% confidence intervals we would need to calculate the probability of goodness of fit. Because of the limited number of data points in combination with the stochastic nature of the system, we do not feel confident in trusting the probability density function (PDF) calculated from experimental data (needed for the 95% confidence intervals). We considered increasing the data, which would take several months, but even doubling the sample size, with the current diversity of clones would not enable us to calculate the probability of goodness of fit of experimental data.
 - b. What we do is the opposite: we estimate PDFs from the simulations and compare our experimental data to these PDFs. (As also mentioned by reviewer in pt 5 below). This gives us likelihood values, but no good way to get actual probabilities of the goodness of fit, and thus no confidence interval. These likelihood values can still be used to select one model over the other, but do not tell us if either model fits reality. We have added a sentence on this in the model section: "The statistical approach used allow us to identify the best model but a combination of limited amount of biological data and high stochasticity prevents us from statistically testing how well each model match the data."

3. *A plot of the experimental distributions of clone size/make up against the "best" parameters for each model would make the model agreement more clear too; ...*
 - a. True, and we did try this, but again because of the high stochasticity and too few data points, the overlay between the simulations of the two models and data is not visually informative. E.g. when we tried making Fig 5D for both models, there was no obvious visual difference, as even for the best parameters difference realization would have somewhat different compositions. Alternatively, if reviewer had in mind plotting data points on top of Suppl. Fig 8E and 8F, again the data points were too few to visually provide any information.
4. *... though the likelihood calculations for the two models will allow the best parameters for each model to be calculated, it is not in and of itself a measure that the models recapitulate the data well.*
 - a. True, this is related to pt 2 above. What we are aiming at estimating is how much one model is better than the other not how well they capture the data. It is obviously best to have a measure of “goodness of fit”, however when we considered doing additional experiments to e.g. double the amount of data points, we realized that this won’t bring us closer to estimating confidence intervals. Again , the main challenge is high clone-to-clone variability.
5. *There are graphs in SFig 8 that appear to show this for different populations/times, but the figure and legend are not clear enough to know whats going on. For example, I cannot see the simulation points in the graph, and I'm not sure why the PDF is calculated from the experiment and compared with simulation rather than the other way around. Intuitively, the PDF calculated from the simulation data should be better sampled and then the experimental observations noted.*
 - a. We apologize that the figure caption was unclear. The PDF is indeed calculated from the simulations. We have edited figure caption to stress that all subpanels of Suppl. Fig. 8 are results of the simulations.
6. *Also, it seems surprising that the PDFs have a "doughnut" shape; is the center of the distribution really as unlikely as points far from the edge of the distribution?*
 - a. We do apologize; the white core of the PDFs was clusters of the white points representing the data. We removed them in the current version, as we don’t think this overlay provided any information.
7. *The authors argue that a time dependent value of f describes the data better than a constant value of f, and make some specific predictions (Fig 5f), though its not clear how this could be validated. I note that 5f isn't referenced in the text, and would recommend to the authors that the prediction is either integrated with the rest of the paper and experimental validation performed or suggested, or the subfigure removed if its not a testable prediction.*
 - a. Thank you for pointing this out. We indeed used the model to make a prediction that we tested experimentally in the next paragraph but it was not very clearly stated. We now clarify this by the following sentence in the text: “According to the model prediction, acinar-committed cells should be undetectable at E9.5, as we have seen, but should be

easily identified by clonal lineage tracing from E11.5 (Fig. 5f).” Fig. 5f was originally referenced in the beginning of the section “Acinar-committed cells are detected from E11.5-12”, the one following the model section.

8. *A minor point on the model annotation; the choice of "q" is unfortunate as the labels on the parameter scan are easily misread as "b".*
 - a. We have fixed this now.
9. *The authors also submitted some code with the recent revision. This is a big improvement from before, but the code is not standard python. From reading through the files it appears to be in a python-notebook type format. The authors need to state how a user can interact with this. To be clear; I believe that the work is reproducible, and that a sufficiently dedicated and experienced researcher could get this working, but the authors need to document it better.*
 - a. In addition to the python-notebook we now provide code in both the standard python format as well as a readme file with the directions on how to run a python-notebook file.
10. *Finally, there are some presentation issues. A lot of the new text has typos e.g. Fig5A, last panel "lenght", page 7 "allowsallowed", page 7 "estimateestimated" page 12 "To estimate a probabilityprobabilty" etc. These should be fixed; it distracts from what is otherwise a nice, clear description of whats going on.*
 - a. We apologize for this, this has now been fixed.
11. *Furthermore, graphs need keys on the graph themselves if they are to be readable.*
 - a. We have added the key labeling the cell types to the Figure 5 and Suppl. Figure 8.

Reviewers' Comments:

Reviewer #4:

Remarks to the Author:

The manuscript has been greatly improved and I'm happy to recommend acceptance. I note that I have been able to run the code but due to time constraints on the review the simulations have not been able to complete; however the code appears sound and is readable.

Minor notes:

I recommend changing the line `get_ipython().magic('matplotlib notebook')` to something else as this prevents the code from running (at least on my machine).

Authors should consider clarifying some of the code comments. Whilst informal comments are fine, I don't think that users will struggle to understand variables with comments like "`# <-- MISTAKE IN CODE: IT WOULD HAVE NO DIFFERENCE!!`" or "`# <-- I CHANGED THIS VALUE SO THE DIST WOULD MATCH THE PICTURE ONE FROM THE NOTEBOOK FROM Anna(?)`"

Still some typos in the manuscript: e.g. page 7 "The statstical approach"